# Balancing land use for conservation, agriculture, and renewable energy

Cameryn Brock [1] ✉, Patrick R. Roehrdanz [1], Tim Beringer[2], Rebecca Chaplin-Kramer [3], Brian J. Enquist[4], Amy E. Frazier [5], Justin A. Johnson [6], Christina M. Kennedy [7], Joseph Kiesecker[8], Ashley E. Larsen [9], Rafael Loyola [10,11], Pablo A. Marquet [12,13,14,15,16], Rachel A. Neugarten [17,18], James R. Oakleaf [8], Anand Roopsind [19], Richard Schuster [20,21], David R. Williams [22], Grace C. Wu [23], Alex Zvoleff [1] & Lee Hannah[1]

Growing demand for food coupled with climate commitments to reduce emissions will result in more land development for agriculture and renewable energy. Simultaneously, conserving land for biodiversity and nature's contributions to people (NCP) is imperative for achieving international climate, sustainable development, and biodiversity goals. Meeting these interconnected objectives requires efficient land allocation across sectors. Here, we present a flexible, multiple-objective framework for strategically allocating land to mitigate threats to biodiversity and NCP under climate change while supporting development. Application of this framework at a global scale through country-level targets shows that if future development is planned without consideration of nature, demands for land could impact nearly 1 million km² of high-priority conservation areas. Multi-sector planning can mitigate potential conflict, reducing carbon loss and species exposure. Our findings underscore the need to conserve critical areas for nature, reduce land demand for food and energy, and intentionally coordinate land use across sectors.

Climate change presents an urgent challenge, with the policy decisions, technological advancements, and sustainable practices we adopt today shaping the future of both human society and the natural environment (IPCC[1]). The path to a net-zero emissions future requires an immediate focus on changes in land use and energy development across sectors: Opportunities for climate change mitigation lie in transforming agricultural systems and diets, implementing natural climate solutions like reforestation and avoided deforestation, and swiftly transitioning to renewable energy sources (Barthelmie & Pryor[2], Creutzig et al.[3], Gielen et al.[4], Griscom et al.[5], Roe et al.[6], and Searchinger et al.[7]). Climate change solutions can benefit and include biodiversity (Pacifici et al.[8], and Griscom et al.[5]); however, without coordinated planning, popular forms of mitigation such as renewable energy development can further exacerbate the ongoing biodiversity crisis (Hernandez et al.[9], Köppel et al.[10], Levin et al.[11], Wu et al.[12], and

Condon et al.[13]). The international community has recognized the urgent need for greater investment in climate and biodiversity through the Kunming-Montreal Global Biodiversity Framework and the Paris Climate Agreement. Global commitments include protecting 30% of the planet's important areas for nature, halting the loss of any additional areas through integrated, participatory spatial planning (CBD[14]), and limiting global warming to well below 2 °C (UNFCCC[15]).

Growing demand for food and energy associated with rising population and per-capita consumption increases the urgency of efficient land use planning. Projections indicate that demand for cropland and energy will grow across a range of future climatic and socio-economic scenarios, driving an expansion of land conversion (Popp et al.[16], Riahi et al.[17], and Boakes et al.[18]). Often overlooked, energy generation constitutes a significant form of land use. Land use for energy generation will increase in the coming decades, with increases

rivaling those associated with agriculture (Johnson et al.[19] and Trainor et al.[20]). Renewable energy sources have large land use requirements, with cropland for bioenergy exceeding other sources in projected land use (both in total area and per unit of power), followed by wind energy, hydro power, and solar energy (Fthenakis & Kim[21]; McDonald et al.[22], Trainor et al.[20], Nøland et al.[23], and Lovering et al.[24]). While large areas will be required for future renewable energy development, renewable sources can continue generating energy over time within their footprint, whereas extractive energy sources such as coal, oil, and gas, must continually open new drilling or mining sites to maintain output, resulting in comparable land use requirements across certain renewable and fossil technologies (Dai et al.[25]). While land use competition from solar and wind projects in both rural areas (Spangler et al.[26], Nilson & Stedman[27], and Rand et al.[28]) and natural habitats (Katzner et al.[29] and Levin et al.[11]) have intensified, siting strategies that avoid conflict (Wu et al.[12]) or increase land use compatibility like agrivoltaics (Swanson et al.[30] and Merheb et al.[31]) have begun to emerge.

Siloed land-use planning could lead to patterns of human infrastructure that harm natural ecosystems and wildlife. To balance increased development demands and climate solutions with minimizing extinctions, it is crucial to address the existing impacts on biodiversity driven by habitat loss and fragmentation and to avoid further loss (Isbell et al.[32], Jaureguiberry et al.[33], Pörtner et al.[34], and Smith et al.[35]). Meeting humans' growing demands while addressing the urgency of biodiversity conservation and climate change mitigation will require strategic, collaborative planning to optimize land use across sectors (Fastré et al.[36] and Kiesecker et al.[37]). Specific attention to the dynamics of shifting climatic suitability for species and crops will be essential to ensure that resources are used effectively and land conserved is suitably located (Hannah et al.[38] and Hannah et al.[39]).

Previous research has examined global geographic priorities for biodiversity, carbon, and other natural assets that deliver key benefits to people (Chaplin-Kramer et al.[40], Jung et al.[41], Neugarten et al.[42], and Strassburg et al.[43]), with priorities for biodiversity including those with consideration of climate change (Hannah et al.[39]) and the intersection with future land uses such as agriculture (Kehoe et al.[44] and Williams et al.[45]) and solar and wind power (Dunnett et al.[46], Kiesecker et al.[47], and Rehbein et al.[48]). These advances explore the components of future planning individually, leaving underexplored the potential for

multi-sector planning to inform the intersection of future conservation, agriculture, and renewable energy. Here, we present a framework for strategically allocating land under climate change to mitigate threats to biodiversity, carbon, and nature's other contributions to people (NCP) while supporting development for agriculture and renewable energy. Our approach builds on previous studies (Hannah et al.[39], Neugarten et al.[42], and Johnson et al.[19]) to offer a spatially-explicit examination of this multifaceted problem and propose solutions through an integrative, cross-sector method. Importantly, our framework allows the exploration of multiple scenarios with relatively low computational power, enabling usage by a broader audience than larger models (e.g., integrated assessment models). In this study, we hypothesize that strategic allocation can be a critical solution to help address the converging challenges facing our ecosystems and development landscapes.

## Results
### Scenario framework

We developed three planning scenarios to evaluate how differing priorities affect optimal land allocation for conservation, agriculture, and renewable energy (Fig. 1). The "Production-First" planning scenario allocates land to meet future demand for development (for energy and food) based on potential yield, siting constraints, and siting feasibility (e.g., distance to roads and powerlines), and the remaining land is then available to be optimally allocated for conservation. The "Nature-First" planning scenario first allocates land for conservation, and the remaining land is allocated for development. The "Multi-Sector" planning scenario allocates land for conservation and development concurrently. This innovative scenario framework allows planners to explicitly examine tradeoffs by facilitating a comparison of areas that would be allocated differently depending on planning priorities.

Land for development is allocated to meet targets for renewable energy (photovoltaic [PV] solar power, concentrated solar power [CSP], wind power, hydroelectric power, and biofuel crops) and food crops. The targets for each region and sector were retrieved from the Shared Socioeconomic Pathways (SSP) database, defined as future sector-specific demand for energy in gigawatts (PV solar power, CSP, hydro power, and onshore wind power) and tonnes of dry matter

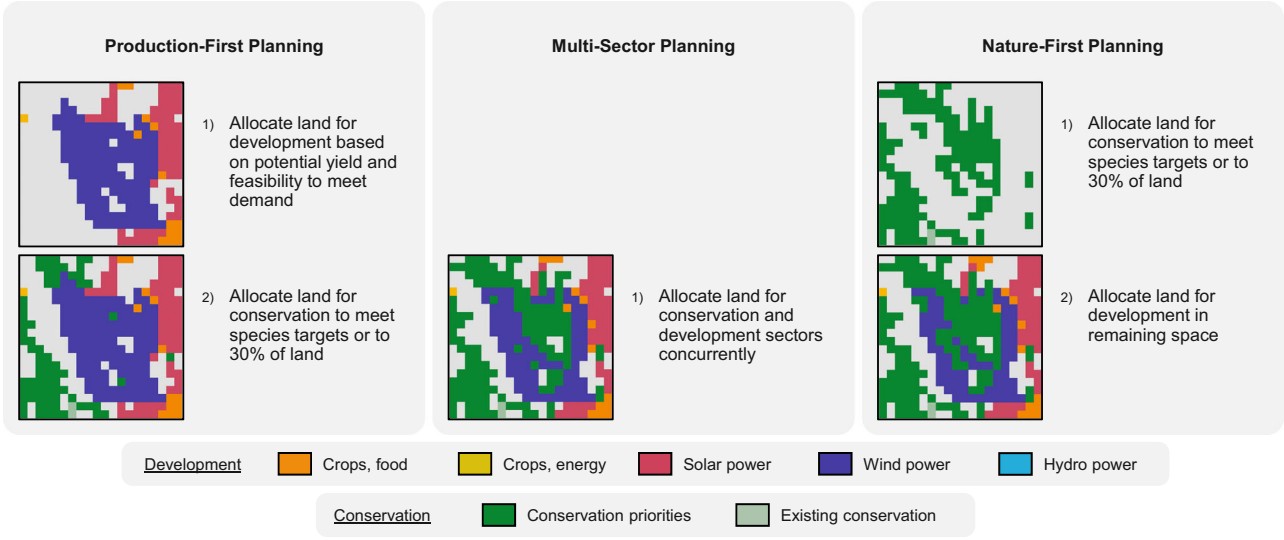

**Fig. 1 | Three planning scenarios for evaluating how differing priorities affect land allocation and the simplified workflow for the spatial optimization approach in each: (left) Production-First, (middle) Multi-Sector, and (right) Nature-First.** The square insets are zoomed-in maps of one example geographic location illustrating different results from the three scenarios. Development includes food crops (orange), energy crops (yellow), solar power (pink), wind power (purple), hydro power (blue) and conservation includes existing conservation (light green) and conservation priorities (dark green). Expansion of multiple sectors in one pixel was not considered.

(biofuel crops and food crops). Projections for food and energy reflect the SSP 1 "Sustainability" narrative for the year 2050, aligned with ambitious global sustainability goals that seek to mitigate climate change and promote responsible consumption (Riahi et al.[17]). The region-level (see regions in Fig. S1) development targets were attributed to each country based on market share (see Methods) to run the analysis at the country level. We assume that additional conservation and development come from the conversion of available land, with available land excluding current dense development (more than 80% human modification, see Methods). We are using a pixel-based approach that allocates each pixel to one sector and does not allow for co-location of multiple sectors (for example, wind turbines in future agricultural lands). This decision is based on current available data and the practical, context-specific considerations that are needed at local scales to co-locate different land uses. Accordingly, our results represent upper bounds on additional land requirements in this scenario; realized outcomes will depend on local technical, agronomic, and biodiversity constraints.

In all scenarios, land for conservation is allocated to meet targets for biodiversity, carbon, and other NCP. For biodiversity, we focus on 7199 terrestrial vertebrate species categorized by the IUCN Red List (IUCN,[49]; BirdLife International[50],) as Near Threatened and threatened (Vulnerable, Endangered, and Critically Endangered). Each species' habitat is established based on the overlap of their present habitat (Area of Habitat; see Methods) and future climatic niche (Hannah et al.[39]). Biodiversity targets were defined at the species level based on global range size (Hanson et al.[51] and Rodrigues et al.[52]). To implement these targets at the country level, the percentage targets were used in combination with the country-level habitat. For example, if a species' global target was to conserve 20% of its habitat, each country with that species would have a conservation target set at 20% of the habitat in that country. Carbon includes vulnerable terrestrial carbon, as described by Noon et al.[53]. NCP build on the ecosystem services concept to recognize the role of culture in the linkage between people and nature (Díaz et al.[54]), operationalized here with the use of spatially-explicit models that combine ecological supply of benefits with the populations they benefit (Chaplin-Kramer et al.[40]), including those for nitrogen retention, sediment retention, coastal risk reduction, pollination, and access to nature. Targets for conserving carbon and NCP were set to 90% of the current levels of each NCP to align with the critical natural assets concept from Chaplin-Kramer et al.[40].

Land allocation for each target is spatially optimized using mixed integer linear programming (ILP; Hanson et al.[55]). ILP combines mathematical modeling with computational algorithms to find the best possible solutions (e.g., where to place a conservation area to represent the most species) given constraints provided (e.g., budget limits or land availability). ILP solvers can outperform other approaches in reaching cost-effective solutions for conservation problems, both in terms of the optimality of solutions and in computational efficiency (Beyer et al.[56] and Schuster et al.[57]). For each scenario, we ran the analyses in two ways: first, constraining the analysis to allocate up to 30% of land per country to conservation (including existing conservation areas), and second, permitting the percentage of land allocated to conservation to be as high as needed to meet individual targets for species, carbon, and NCP, which is often higher than 30% (although it can be lower).

Our framework provides several ways to account for climate change in land-use planning. First, conservation priority areas are determined at the species level based on the overlap between each species' current habitat and their projected future climatic suitability. By accounting for the projected changes in species' climatic suitability, conservation initiatives can adapt and allocate resources strategically, facilitating vulnerable species' long-term survival and improving the overall effectiveness of conservation efforts (Hannah et al.[39]). Second, our allocation for agricultural land is based on future, rather than solely current, climatic suitability for crops. Growing conditions for crops are shifting, providing both new opportunities for agriculture and reducing the suitability of existing cropland, resulting in environmental tradeoffs that are important to consider (Hannah et al.[38]). Lastly, we base our projections for climatic shifts (related to species and crop suitability) on a moderate climate change scenario, while our estimates of demand for renewable energy are based on what is needed to meet a low emissions scenario. This distinction allows planners to prepare for the worst impacts of climate change while working toward mitigating those impacts in a way that aligns with global emissions reduction targets. Our decision to use a moderate emissions scenario for yields also aims to account for uncertainties in climate sensitivity and other non-linear processes in the climate system that the climate models did not account for.

## Land allocation for conservation, agriculture, and renewable energy

The ability to meet multiple objectives depends on the degree of coordination across sectors and the scale at which conservation objectives are implemented. In the Multi-Sector planning scenario with national 30% conservation targets, 2.0 million km$^2$ of currently undeveloped land (2% of total land area, equivalent to the land area of Mexico) is converted to agriculture globally by 2050, with 56% dedicated to food crops and 44% to biofuel crops. An additional 4.5 million km$^2$ (3% of land) is allocated for renewable energy generation: 76% for wind power (including spacing between turbines); 19% for solar power, and 5% for hydro power. In addition to existing conservation areas, 20.4 million km$^2$ (15% of land) is allocated for conservation to meet 30% of land conserved per country (Fig. 2; see Figs. S2 and S3 for other scenarios). Aligned with SSP projections, regions with the largest allocation for development include eastern and southern Asia, most notably India, China, and South Korea, northern coastal Africa in Algeria and Morocco; the central United States; and western Europe. Land allocation for all countries and scenarios can be found in Supplementary Data 1.

Our results indicate that if the increase in development to meet future demand relies solely on additional land conversion without co-locating sectors (e.g., wind energy on agricultural land), it is not possible to meet the country-level targets for conservation and development needs. Conservation and development targets met varied by scenario, with the Nature-First planning scenario generally more effective in meeting conservation targets, Production-First more effective at meeting development targets, and Multi-Sector falling in between (Fig. 3A).

The extent to which conservation targets were achieved primarily depended on whether land allocations were constrained to meet a 30% target for conservation and, to a lesser extent, on the planning scenario. For biodiversity, targets are only met for slightly less than half of the species when constrained to 30% land conservation. Removing the constraint led to targets being achieved for nearly all species in Nature-First and Multi-Sector and 88% in Production-First. For NCP, when constrained to 30% of land for conservation, targets are met in 3% of countries in the Production-First planning scenario and 9% in Multi-Sector and Nature-First. Removing the 30% land conservation constraint allows NCP targets to be met in 96% of countries in the Nature-First planning scenario, 84% in Multi-Sector, and 62% in Production-First. Without the area-based constraint, we found that 56% of all land is needed for conservation globally, which is 41% above what is already conserved (Fig. S5). Nearly all countries require more than 30% of land conserved to meet conservation targets (Fig. S6 and see Supplementary Data 1), with variation across SSP regions: Latin America (LAM) required 69%, Organization for Economic Co-operation and Development countries (OECD) 58%, Asia (ASIA) 57%, Middle East and Africa (MAF) 52%, and the countries from reforming economies of Eastern Europe and the former Soviet Union (REF) 48%.

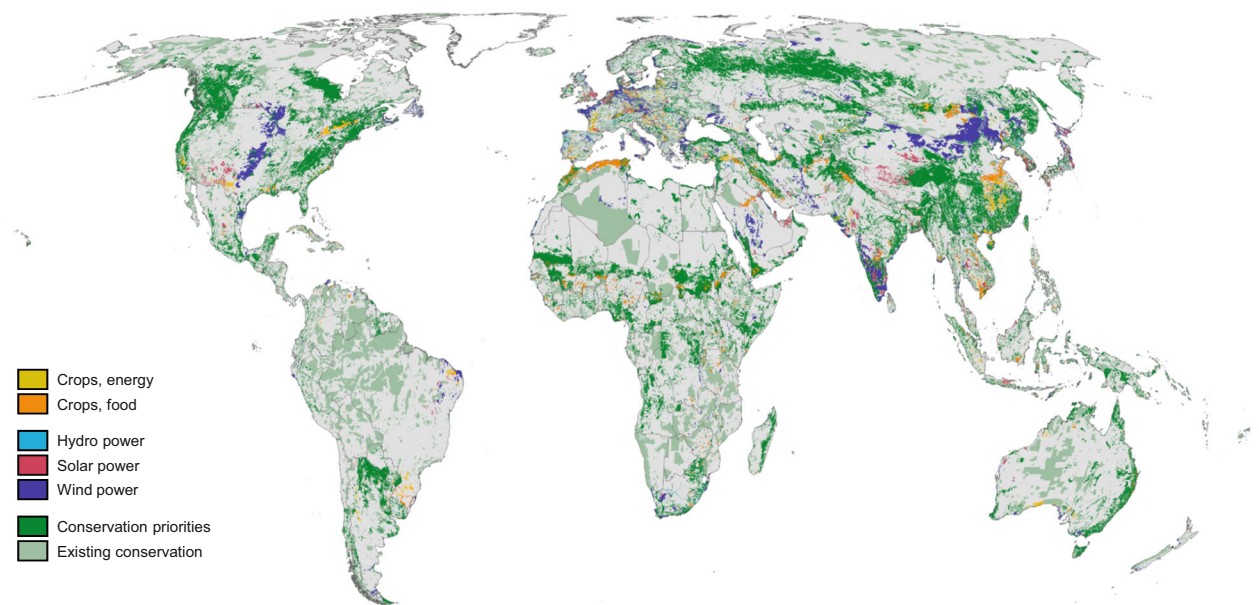

**Fig. 2 | Additional land allocated for conservation and development in the Multi-Sector planning scenario.** Development includes food crops (orange), energy crops (yellow), solar power (pink), wind power (purple), and hydro power (blue) and conservation includes existing conservation (light green) and conservation priorities (dark green). Each country has 30% of its land allocated to conservation, including existing conservation areas. The area-based threshold of 30% is based on global commitments and does not necessarily reflect the amount needed to meet conservation targets. Demand is based on 2050 projections. Existing conservation areas include protected areas and other effective conservation measures (UNEP-WCMC & IUCN[49,114]). Country boundary data are from Natural Earth (www.naturalearthdata.com).

The percentage of development targets met was 94% in the Production-First planning scenario, 74% in Nature-First, and 87% in Multi-Sector. Production targets achieved also varied by sector. For example, in the Nature-First scenario, approximately half of the hydro power and wind power targets were achieved, whereas over 80% of targets were achieved for energy crops, food crops, and solar energy. Land efficiency for development, meaning the amount of land allocated per tonne of crop dry matter or gigawatt of energy generated, varied by both planning scenario and by sector. For example, hydro power land efficiency ranged more than two-fold, from 64 km²/gigawatt (GW) in Production-First to 148 km²/GW in Nature-First. This finding suggests that hydro power may be difficult to efficiently site when avoiding conservation areas because areas suitable for hydro power are more constrained. Conversely, solar power land efficiency was not as dependent on the planning scenario, ranging only between 34–36 km²/GW across all scenarios. This result aligns with previous findings that solar power is relatively land efficient (Bolinger & Bolinger[58] and Fthenakis & Kim[21]) and flexible in siting (Baruch-Mordo et al.[59]).

Our multi-objective optimization framework can enable decision-makers to address tradeoffs in the case of potentially conflicting objectives explicitly. In our global application, we examined differences in land efficiency (Fig. 3B), and the extent to which species and carbon were exposed to the development footprint (i.e., the spatial overlap of carbon or over 10% of a species' habitat [Fig. 3C, D]). Per unit of development, the Production-First scenario was the most land efficient, yet it consistently led to greater exposure for species and NCPs. This tradeoff demonstrates spatial overlap between the areas most suitable for development and those for conservation, consistent with previous research (Neugarten et al.[42]). Total land allocated per sector, which may be indicative of cost, did not vary substantially across scenarios: mean land allocated ranged from 5134 to 5172 km² per target met for conservation and from 7154 to 8254 km² for production (Table S1, and see Supplementary Data 1 for country-level land allocation). Given varying costs and priorities, land-use planners can benefit from explicit consideration of tradeoffs between production

output and conservation goals in conflict areas. To exemplify a potential method for doing so, we have assessed the Pareto frontier reflecting countries' individual optimizations to provide a comparison of the severity of tradeoffs across scenarios (Figs. S6 and 7).

Under the Production-First planning scenario, 674 Near Threatened and threatened species (including 214 endangered species) were exposed to development, compared to 572 (172 endangered) in Multi-Sector and 510 (148 endangered) in Nature-First (Fig. 3C). The relative size of impact varied by species and scenario, with development overlapping, on average, 4.3% of each endangered species habitat in the Nature-First planning scenario compared to 6.0% in Production-First (see the range of species-specific impacts in Fig. S9). Carbon exposure also varied considerably: in Production-First, development overlapped with 21,462 megatonnes (Mt) of carbon, compared to 17,392 Mt in Multi-Sector, and 17,060 Mt in Nature-First. Because the difference across planning scenarios is likely partially due to differences in development targets met, species and carbon exposed are reported per unit of development in Fig. 3B–D.

Independent, single-use allocations for PV solar and wind energy indicate measurable overlap with land allocated for crops for food and energy. Globally, 2.7% of land allocated for PV solar energy overlapped with land allocated for total crop expansion (or 1.1% of crop expansion overlapped with that for PV solar energy), primarily in Europe and India (Fig. S10), and 2.1% of land allocated for wind energy overlapped with that for crop expansion (or 3.5% of crop expansion overlapped with that for wind), primarily in Europe and China (Fig. S11). These results highlight where, if technically feasible, co-location could reduce the need for additional land conversion.

## Potential conflict between important areas for conservation and development

To identify areas of potential conflict between conservation and development, we overlaid land allocation for development from the Production-First planning scenario and land allocation for conservation from the Nature-First scenario (Fig. 4). This revealed several regions where important areas for development coincide with those

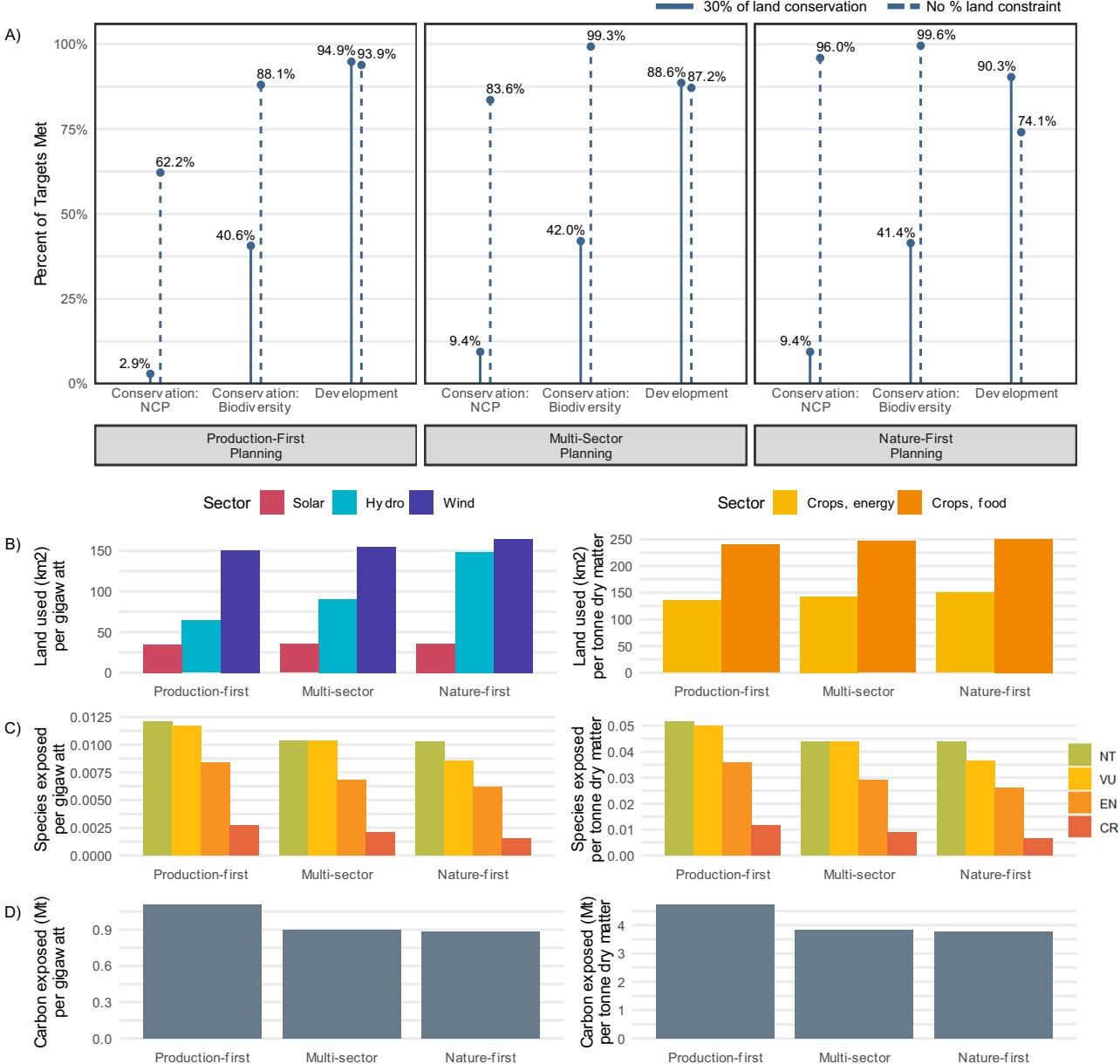

**Fig. 3 | Evaluation of targets met, additional land used, and species and carbon exposed per scenario. A** Percent of targets met for conservation and development features. All land allocation is performed at the country level, and whether the targets are met is assessed at the country level for Nature's Contribution to People (NCP) and development and at the species level for biodiversity. The solid line reflects allocating 30% of each country to conservation, and the dotted line reflects conservation allocation without a 30% constraint. **B** Land allocated per unit of development (gigawatts for hydro power [blue], solar power [pink], and wind power [purple]; tonne dry matter for energy [yellow] and food [orange] crops).

**C** Species exposed to development per unit of development. Species exposure is at least 10% of its range intersecting with the development footprint. Colors reflect the IUCN Red List Category of each species: Near Threatened (NT; green), Vulnerable (VU; yellow), Endangered (EN; orange), and Critically Endangered (CR; red). **D** Carbon exposed to development per unit of development and reported in megatonnes. Due to differences in units, (**B**–**D**) are split into hydro, solar, and wind energy in the left column and crops for energy and food in the right column. Demand is based on 2050 projections.

critical for future conservation, including eastern and southern Asia, western Europe, and northern Africa (Fig. 4 insets). The ASIA and OECD regions had the largest area of overlap at 439 K km² and 270 K km², primarily driven by wind overlapping with conservation, followed by the MAF region at 229 K km², primarily driven by food crops overlapping with conservation (Fig. S12). MAF had the highest proportion of development that overlapped with the 30% conservation allocation at 16%, followed by ASIA and OECD at 14% and 12%. Countries with the most significant areas of overlap for conservation and development priorities included China (189 K km²), India (106 K km²), and the United

States (75 K km²) (Fig. S13). While potential conflict areas signal that the land is compatible for multiple land uses, they do not necessarily signal that the identified land uses themselves are compatible. Existing empirical work suggests that biodiversity will fare poorly on developed lands (Kumara et al.[60] and Conkling et al.[61]) and that targeting low conflict areas, particularly for energy development, can result in better outcomes (Kiesecker et al.[62]).

While our framework would allow development to be sited in most areas with potential food or energy yield (Oakleaf et al.[63]), the model is designed to avoid siting development in current conservation

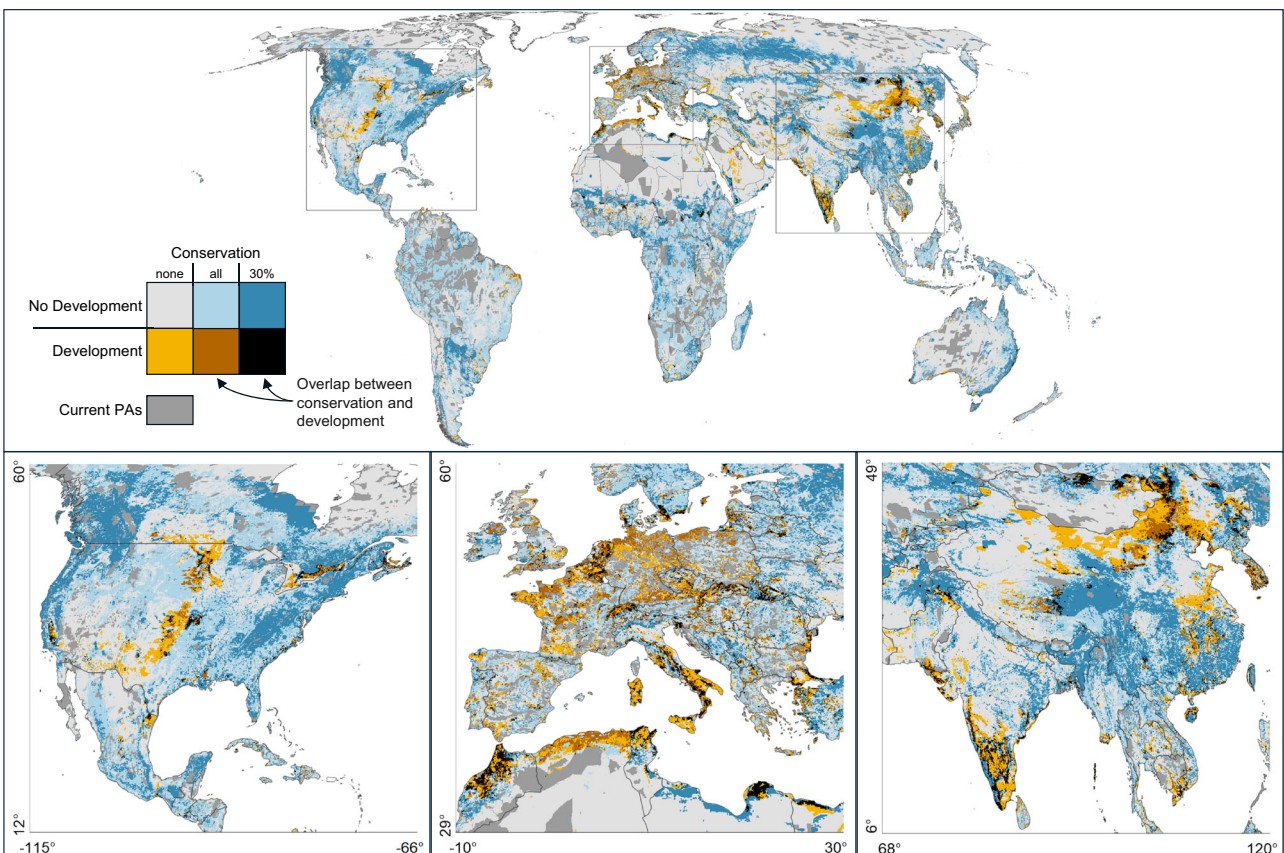

**Fig. 4 | Area of potential conflict between additional land allocated to conservation under the Nature-First planning scenario and development under the Production-First scenario.** For conservation, light blue reflects the area needed to get as close as possible to meeting all country-level nature targets. In contrast, the dark blue is the optimal area for conservation when constrained to 30% of land. Yellow reflects land that was allocated for development. Any overlap of land allocated for development and conservation is in the brown and black categories, depending on whether the development overlapped with the conservation areas needed to get as close as possible to meeting nature targets (brown) or conservation areas constrained to 30% of land (black). Overlap of development and current conservation areas (PAs) is also included in the black category (see Fig. S14). Demand is based on 2050 projections. Existing conservation areas (gray) include protected areas and other effective conservation measures (UNEP-WCMC & IUCN[49,114]). Country boundary data are from Natural Earth (www.naturalearthdata.com).

areas by imposing an additional cost (see Methods). Despite the higher cost, development was sometimes allocated within current conservation areas (Fig. S14), signaling potential risk to biodiversity and NCP in those areas. Countries with the most notable overlap between land allocated for future development and current conservation areas included Mongolia (7200 km²), China (5150 km²), Japan (5025 km²), Italy (4975 km²), and Niger (4150 km²).

Overlap with conservation varied by development sector (Fig. 5). While hydro power has the smallest total footprint by area, it was the most likely to overlap with land allocated for conservation, likely because areas that are riparian and of high topographic variability have high biodiversity and provide significant levels of NCP in the form of water quality and sediment regulation. Further, the Multi-Sector planning scenario was less effective in mitigating conflict for hydro power than for other sectors: 18% of hydro power development overlapped with conservation areas in Multi-Sector compared to less than 7% in all other sectors. In Production-First, land allocated for hydro power had the highest overlap with conservation areas (35%), followed by crops for food (23%) and energy (18%), solar power (14%), and wind power (13%). The higher the overlap between cropland and conservation may be because cropland relies on higher land productivity (e.g., soil quality), which is not necessary for solar and wind power. Overlap between solar power and conservation priorities was most effectively reduced with the Multi-Sector planning scenario. In Production-First, the 14% of development overlapping with conservation dropped to 2% in Multi-Sector.

## Discussion

An improved strategy for siting and managing productive landscapes is needed to avoid further habitat destruction, otherwise, energy and food security will come at the cost of biodiversity, carbon, and NCP. Meeting the ambitious targets outlined in the Kunming-Montreal Global Biodiversity Framework, the Paris Climate Agreement, and the UN Sustainable Development Goals will require rapid expansion of conservation and renewable energy generation – potentially driving conflicting demands for land. Indeed, renewable energy development has already led to deforestation and forest degradation in some regions, threatening biodiversity and our ability to mitigate climate change (Fargione et al.[64], Ortiz et al.[65], Keles et al.[66], Rehling et al., and Zhang et al.[67]). Avoiding these conflicts will require an integrative spatial planning approach and the ability to integrate multiple objectives and assess tradeoffs. We have presented a multi-sector framework that seeks to address the critical challenges of land allocation under climate change to mitigate threats to conservation features while supporting food and renewable energy development. Our framework emphasizes the flexible nature of locating renewable energy development, a characteristic that sourcing for conventional energy lacks, offering a promising opportunity to avoid harm to natural areas through careful planning and collaboration. Our framework for strategic, multi-sector planning aims to empower land use planners to find synergies across sectors and transcend the traditional trade-offs encountered in siloed approaches.

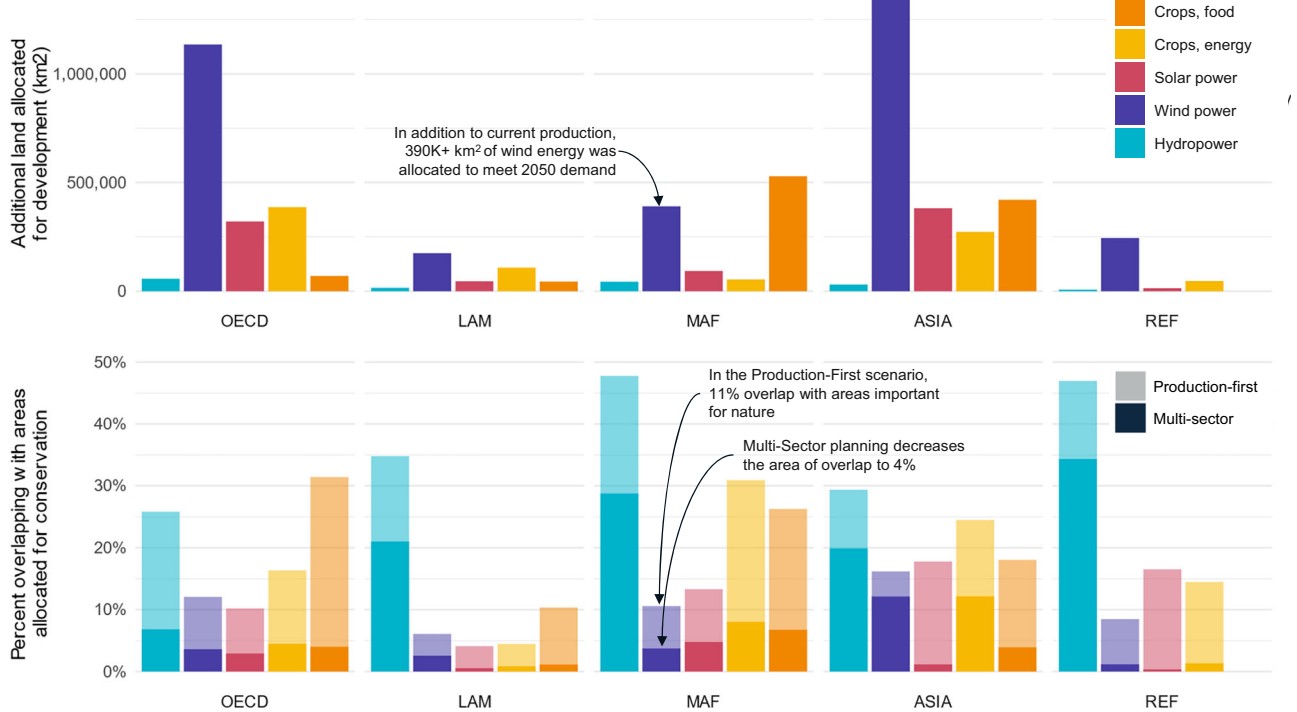

**Fig. 5 | Additional land allocated for development and the proportion overlapping with areas allocated for conservation.** The top panel shows the total area of additional land allocation in the Production-First planning scenario per region for food crops (orange), energy crops (yellow), solar power (pink), wind power (purple) and hydro power (blue). The bottom panel shows the percent of the total area of additional land allocated that overlaps with additional areas allocated for conservation in the Nature-First planning scenario or areas of potential conflict between conservation and development, both in the Production-First (lighter color) and Multi-Sector (darker color) planning scenarios. Regions align with the Shared Socioeconomic Pathway (SSP) framework (Riahi et al.[17]), which includes Asia (ASIA), Latin America (LAM), Middle East and Africa (MAF), Organization for Economic Co-operation and Development countries (OECD), and the countries from reforming economies of Eastern Europe and the former Soviet Union (REF) as shown in Fig. S1. Demand is based on 2050 projections.

While our methodological approach aligns with previous research in using the SSP database to allocate land for future food and energy production (e.g., Popp et al.[16] and Johnson et al.[19]), our aim is not to predict which areas will be allocated to specific land uses in the future. Instead, we demonstrate an approach to including multiple objectives in land-use planning as a critical solution in the face of the ongoing biodiversity and climate change crises. Combining this approach with country-level data, local and Indigenous perspectives (Heiner et al.[68] and Kennedy et al.[69]), ongoing efforts to limit land conversion (such as through zero-deforestation commitments; Garrett et al.[70]), and commitments for restoration (e.g., Fagan et al.[71]) has the potential to minimize conflicts between conservation and development planning when compared to siloed approaches.

Our analysis suggests that if increased demand for food and energy is met without co-locating sectors, it is not possible to meet conservation and development goals. This key result highlights the need to conserve critical conservation areas, curtail demand proactively, and use land more efficiently. For example, closing agricultural yield gaps could significantly increase the efficiency of agricultural land use (Folberth et al.[72] and Mauser et al.[73]). Multi-functional landscapes could also potentially limit the need for additional land conversion. For example, solar and wind power can be co-located with each other (Deshmukh et al.[74]) and with agriculture (Maguire et al.[75], Ravi et al.[76], and Miskin et al.[77]), with the opportunity to improve yields (Hernandez et al.[78]). Similarly, siting solar on rooftops, within urban landscapes, or on previously converted lands could make sizable contributions to overall clean electricity supply (Adeh et al.[79], Baruch-Mordo et al.[59], Battersby[80], Hernandez et al.[81], and Joshi et al.[82]). However, there is less research available addressing whether co-located land uses between development and natural lands support biodiversity and NCP. The limited literature that is available on biodiversity outcomes in shared-use landscapes suggests that biodiversity can fare poorly (Rehling et al.[83] and Phalan et al.[84]). For example, while development for wind energy may avoid total conversion of land, this same pattern can drive habitat fragmentation and associated species impacts when sited in natural areas (Rehling et al.[83] and Kiesecker et al.[62]). Determining where and how co-location is appropriate requires site-specific assessments that consider ecological, technical, and socio-economic factors and given current available knowledge on this topic, its viability and outcomes are best evaluated at the local scale. Due to these uncertainties and tradeoffs, and a lack of global data, we did not explicitly account for the potential of co-locating future land uses within planning units.

There are also options for improving the environmental outcomes of development, which will vary based on sector and management practices. Converting natural ecosystems for development has significant environmental consequences, such as the introduction of pollutants, soil erosion, water use, species mortality, and many issues related to land clearing, such as habitat loss and fragmentation (Dhar et al.[85]). While for many sensitive species, agriculture and energy landscapes may not be able to contribute to species-level targets, for other species, improving management practices can mitigate harm (Kremen & Merenlender[86]). For example, the land under solar panels can be revegetated to improve outcomes for biodiversity and soil, management and design of wind turbines can be improved to limit bird and bat mortality, and renewable energy projects can be more strategically located to avoid critical natural areas (Dhar et al.[85], Wellig et al.[87], and Wu et al.[12]).

Our framework includes several additional important analytical decisions and limitations that should be considered. (1) Shifting climate patterns such as irradiance, temperature, and wind are likely to affect optimal locations for renewable energies (Gernaat et al.[88]) and

processes such as nutrient recycling and ecosystem turnover (Weis-kopf et al.[89]). The severity of climate impacts will also shift patterns of agricultural suitability, altering crop allocation. (2) Future technology development and deployment are uncertain, which can lead to over-estimates (e.g., if technologies improve efficiencies) or underestimates (e.g., if energy storage capacity is stagnant), shifting the location and amount of land needed to meet development targets. (3) Our scope does not include all sectors that may drive land conversion in the next 25 years, including some that may have a substantial impact, such as forestry (Mishra et al.[90]) and urban development (Chen et al.[91]). (4) We could not consider important conservation features such as many taxa, including insects, plants, and freshwater and marine biodiversity; connectivity and other ecological and evolutionary processes; and all available NCP. Thus, our conservation targets are likely an under-representation of the total land area required to achieve biodiversity and climate goals as well as maintain flows of NCP that support human life and livelihoods. (5) The choice of spatial resolution will influence results (Arponen et al.[92]); for example, Neugarten et al.[42] found that more land area was required to achieve NCP targets at coarser reso-lutions. Our framework is deliberately flexible to handle variable spa-tial scale and resolution depending on each planner's unique context. (6) Our choice of SSP 1 for land demand aligns with the most ambitious global sustainability goals, which may result in more land allocated for renewable energy and less land allocated for food crops compared to other scenarios.

We recommend that planners carefully consider the scale at which they implement this framework, as the results can be sensitive to scale. While targeting areas of global importance is essential for conservation planning (Shen et al.[93]), our assessments are at the country level to align with the scale at which conservation policies are often imple-mented and to more realistically allocate renewable energy produc-tion. While we based our scenarios on projections of production (rather than demand), the country-level analysis may introduce unrealistic constraints as countries can externalize their food and energy requirements, and we cannot account for many factors that will vary by country. For this reason, we recognize that more locally cali-brated models may yield different results due to their ability to incorporate finer-scale drivers and pressures. For example, recent studies focused on the Gran Chaco project higher rates of deforesta-tion and associated conservation conflict that our analysis suggests (Baumann et al.[94] and Piquer-Rodríguez et al.[95]), highlighting the importance of integrating local land-use dynamics into planning efforts where possible. Allocating demand at the sub-regional level would drive different results as well. In the United States, for example, state-level politics drive the spatial patterns of future renewable energy development (Bromley-Trujillo & Holman[96]) in a way that may not be directly proportional to the resource potential. Nonetheless, such variables were not possible to consider in the model at our global- or country-level scale.

Given these and other constraints when using global data for analyses, we do not recommend using the results from our global analysis directly for local land-use planning. Instead, the global results are intended to identify priority areas, trends, and connections, while landscape-level analyses that inform coordinated and local actions should ultimately be undertaken to advance sustainable land use planning (Frazier[97]). The purpose of this work is therefore not to definitively state where land expansion will occur but rather to high-light that conflicts are possible, to demonstrate that taking a multi-sectoral approach is likely to avoid these conflicts better than focusing on one sector alone, and to provide a solid basis for more nuanced discussions at local and regional scales. Results can be used by con-servation planners to safeguard against climate change impacts, by developers to avoid essential conservation areas, and by government entities to incentivize low-impact development. Our global framework can be used in combination with location-specific data and knowledge and in partnership with Indigenous peoples and local communities to integrate knowledge across scales (Johnson et al.[98] and Chaplin-Kramer et al.[99]) and identify appropriate interventions that align with national and global goals.

## Methods

### Overview

We used integer linear programming with the spatial conservation prioritization R package, "prioritizr" (Hanson et al.[55]) and Gurobi optimization software (version 10) (Gurobi Optimization, LLC.[100]) to strategically allocate land for conservation and six development sec-tors: photovoltaic (PV) solar power, concentrated solar power (CSP), onshore wind power, hydro power, crops for energy (Oakleaf et al.[63]), and crops for food (Fischer et al.[101]). By identifying areas with the greatest conservation value and those with high development poten-tial, we determined the allocation of land use that benefits conserva-tion and food and energy development. All prioritizations were solved to within 1% of the optimal solution.

All spatial raster data were prepared at 5 km using the R package, "terra" (Hijmans[102]). For categorical data, resampling and aggregation were based on nearest neighbor (mode), and for continuous data, aggregation was based on bilinear interpolation (mean). Data resolu-tion affects the outcomes of prioritization (Arponen et al.[92]); we aimed to strike a balance between computational efficiency, the resolution of available data, and the resolution at which we felt it was responsible to interpret future climatic niche data.

We performed prioritization analyses for development and con-servation using the data and parameters detailed in the following sections and in Table S2. All prioritizations were performed at the country level to more realistically allocate renewable energy produc-tion and align with the scale at which conservation policies are often implemented. Given our aim to evaluate the amount of land needed for development without utilization of highly modified landscapes, we omitted land that has existing intensive human activities from the land that was available to the prioritization algorithms. To do this, in all prioritizations, we used the human modification (HM) dataset (Kennedy et al.[103]) to define available land. HM quantifies the propor-tion of each square km modified for human settlement, production, and transportation, with values ranging from unmodified at 0 to completely modified at 1. After aggregation to 5 km, we excluded areas with a more than 80% degree of human modification (aligned with methods in Johnson et al.[19]). For some problems, it was not possible to meet development and/or conservation targets within the available land (7 of 160 countries). In cases that were verified, the potential for development was the limiting factor. To address this, we first attempted to meet 100% of targets, but if that failed, the development targets were incrementally decreased until we could successfully solve the problem. All solutions were evaluated against the original, ideal targets post-hoc to understand what portion could be possibly met, meaning the results account for the cases in which targets could not be met. We considered promoting spatial "clustering" of the solutions with prioritizr's option for a "boundary penalty" but opted not use this approach due to the high sensitivity of the results to the value selected and the lack of a standardized approach for determining a consistent boundary penalty value across country-level prioritizations. Further, the boundary penalty approach can lead to solutions that are highly dissimilar to others and may result in solutions that don't effectively provide functional connectivity (Hanson et al.[104]).

All data were cropped and masked to each country using boundaries from Natural Earth data (www.naturalearthdata.com) using the R package, "rnaturalearth" (Massicotte & South[105]). Country names between the SSP regions and Natural Earth were crosschecked and aligned manually, which, for some countries with contested boundaries, included combining Natural Earth borders to reflect the countries reported in the database. Once the spatial data were

prepared, all layers were converted into sparse matrices to facilitate faster and more efficient prioritizations.

All data analysis and non-spatial visualizations were completed with R (version 4.1.2; R Core Team[106]) and final maps were created with R and ArcGIS Pro (version 3.0.2).

## Scenarios

We developed three planning scenarios to represent different priorities in land-use planning (Fig. 1). The first is "Production-First", in which land is allocated for food and energy development concurrently based on potential yield and feasibility to meet future demand in one prioritizr analysis (or "problem"), and then the remaining land is allocated for conservation to meet targets for biodiversity and Nature's Contributions to People (NCP) in a second problem. Technically, we did this with a "locked out" constraint, meaning that the land from the first problem's solution was not available in the second problem. The second planning scenario is "Nature-First", in which the problems are completed in the opposite direction: first, land is allocated for conservation to meet as many targets as possible, and then that solution is locked out from the subsequent development problem. Finally, the third planning scenario is "Multi-Sector", in which all conservation and development sectors are allocated concurrently in one prioritizr problem. This was completed technically using prioritizr "management zones", in which each sector is a separate zone.

For each planning scenario in each country, we performed the prioritization in two ways: (1) to get as close to meeting conservation targets as possible within 30% of land area and (2) within all land area. 30% was selected as a threshold based on recent country-level commitments aiming to align with the global 30 × 30 goal outlined in the Kunming–Montreal Global Biodiversity Framework. In the analyses for which the conservation area is constrained to 30%, a minimum set objective is used for all problems, meaning that the cost of the solution (detailed below) is minimized while ensuring that all targets are met. Because all conservation targets often could not be met within 30% of land area, this was done by incrementally increasing the percentage of the targets aimed to be met until 30% of land area is met. In the analyses for which there was no constraint, a minimum set objective is used for the conservation problem in the Nature-First planning scenario and the development problem in Production-First. In Multi-Sector, conservation in Production-First, and development in Nature-First, a minimum shortfall objective is used with a budget that is larger than the cost of the solution, simulating an infinite budget. This objective minimizes the overall shortfall for as many targets as possible and allows us to evaluate what targets can be met in all remaining land after the land was optimally selected for one sector.

## Prioritizations for development

We performed the development prioritization to identify areas for agriculture and renewable energy development for the six sectors. These sectors are allocated concurrently (with each other) using prioritizr's management zones. For each energy sector (including bioenergy crops), the features used were sector-level potential yields from Oakleaf et al.[63]. For food crops, potential yield is calculated as the maximum potential yield under a 2050 RCP 6.0 scenario (Fischer et al[101]) across four food crops that occupy the most land globally: maize, rice, soy, and wheat (FAOSTAT[107]). We base our projections for climatic shifts on the RCP 6.0 scenario, versus our demand for food and renewable energy are based on what is needed to meet a low emissions scenario. This distinction allows planners to prepare for the worst impacts of climate change while working toward mitigating those impacts in a way that aligns with global emissions reductions targets.

The targets for each sector were retrieved from the SSP database. Food crops are those for human consumption; land for pasture is not considered as a development category. The SSP database provides demand at the regional level, which we allocated to each country based on their projected market share. For wind and solar power sectors, the projected market share was calculated from Jacobson et al.[108] projections. Jacobson et al.[108] does not include projections for all countries or sectors; thus, the market share for countries not considered in their analyses and for hydro power were calculated based on current market share (IRENA[109]). Country-level market share for cropland is calculated from change in cropland between 2050 and 2020 in an SSP 1 scenario (Chen et al.[110]). The projected market share for cropland was used for bioenergy as well, as Chen et al.[110] do not differentiate whether cropland is used for food or energy crops.

The cost of allocating land for development includes three equally weighted components. Two layers are used in all scenarios: a ubiquitous land cost (value = 1) and feasibility (0–1), defined as the inverse of the sector-specific development potential index (DPI) (Oakleaf et al.[63]). The DPI is a global suitability index at 1-km resolution (with standardized values ranging from 0 to 1) that indicates the potential for development based on spatially-explicit available resources, siting constraints (e.g., land cover, slope), and siting feasibility (e.g., major roads, rails, ports, powerlines, access to demand centers). The third cost layer varies by scenario. While in the Nature-First and Multi-Sector planning scenarios, current conservation areas are excluded from available land ("locked out"), in Production-First, there is an additional cost for current conservation areas (value = 1) (UNEP-WCMC and IUCN[49]) to reflect the economic, political, and social costs of developing in conservation areas and to allow the results to reveal potential risk to those areas. In Nature-First and Multi-Sector, the third cost layer for development reflects value for conservation: this cost reflects the biodiversity, carbon, and NCP features data used in this analysis, each weighted equally. Specifically, the cost layer includes range-size rarity of the overlapping areas between terrestrial vertebrate species' current habitats and future climatic suitabilities (Hannah et al.[39]), vulnerable carbon (Noon et al.[53]), and a sum of five layers of NCP (Chaplin-Kramer et al.[40]). More details on the natural features data are below.

## Prioritizations for conservation

We performed the conservation prioritization to allocate land to meet targets for biodiversity and NCP. We performed the prioritizations in two ways for each planning scenario: (1) to meet feature targets and (2) to meet the greatest proportion of targets possible within 30% of land within each country. To reach 30% within the prioritizr problem, we lowered all the conservation feature targets to 10% of their original target, then increased the targets by percentage increments until 30% of land area was met.

Current conservation areas were compiled from the World Database on Protected Areas (WDPA) and the World Database on Other Effective Area-Based Conservation Measures (WDOECM) (UNEP-WCMC & IUCN[49]). The data were prepared for analysis using the R package, "wdpar" (Hanson[111] and Hanson et al.[51]), then the shapefile data were rasterized to align with all other layers as binary based on the majority of a pixel. The conservation prioritizations are always built on current conservation areas, meaning that when the 30% constraint was used, if 30% or more of land was already protected, no additional land was allocated for conservation.

Natural features data included those for biodiversity, carbon, and NCP. To map biodiversity distributions, we utilized current habitat maps for 7199 Near Threatened and threatened (Vulnerable, Endangered, and Critically Endangered) species, including 2395 birds, 1478 mammals, 2259 amphibians, and 1067 reptiles, and created layers that only included where there was overlap between their current habitat (IUCN,[49]; BirdLife International[50],) with their future climatic suitability (Hannah et al.[39]). Biodiversity is represented as the species-level overlap between "Area of Habitat" (AOH; IUCN, 2020; BirdLife International[50],) and future climatic suitability in an SSP 3, RCP 7.0

2050 scenario (Hannah et al.[39]). This means only the portions of the AOH that remain within a species' projected climatic niche under the SSP 3, RCP 7.0 scenario are considered, an approach that conservatively identifies climate refugia while allowing for limited movement under climate change. Although it would be ideal to align with the RCP 6.0 scenario used for food crops, this scenario was not available. AOH is derived from Red List expert range maps that have been filtered for suitable habitat based on up-to-date satellite data and known elevation limits (Brooks et al.[112]). AOH data used here are based on IUCN and BirdLife International spatial data for terrestrial mammals, birds, reptiles, and amphibians. They are filtered for habitat using a previously described land cover to species habitat table (Santini et al.[113]) and ESA land cover classes for 2018. Spatial data presence codes "Extant" (1), "Extinct" (5) and origin codes "Native" (1), "Reintroduced" (2), and "Assisted Colonization" (6) were used to define each species' Red List range used to produce the AOH. For species that did not have an overlap between their current AOH and their future climatic suitability, their current AOH was used.

Carbon is represented by vulnerable terrestrial carbon storage, which represents the spatially explicit, above-ground and below-ground carbon that could be lost in a typical disturbance event (Noon et al.[53]). NCP features included six data layers that were most relevant to area-based conservation efforts and were publicly available from Chaplin-Kramer et al.[40]: nitrogen retention for water quality regulation, sediment retention for water quality regulation, coastal risk reduction, pollinator habitat sufficiency for pollination-dependent crops, and access to nature (urban and rural). The NCP data includes those "realized" by beneficiaries, including people downstream of habitats providing water quality benefits, populations protected from storm surge, or people within a certain travel time of natural areas.

**Potential conflict**
To generate the potential conflict map (Fig. 4), we compared land allocation outcomes across four sets of results: (1) Production-First with the 30% conservation target (Fig. S2), (2) Nature-First with a 30% conservation target (Fig. S3), (3) Production-First with no conservation constraint (Fig. S4), and (4) Nature-First with no conservation constraint (Fig. S5). Conflict areas represent planning units where the allocation of land diverged across these sets of results, reflecting areas where there may be competing demands between development and conservation.

**Comparison of independent allocations for select development**
To assess the potential for co-location within our single-use framework, we independently prioritized land for crops (food and energy), PV solar energy, and wind energy. We used the same data, constraints, and cost structure as the main analysis, but created independent vectors for each sector's data rather than utilizing prioritizr's management zones. We then analyzed the overlap between the results for (1) land for crops and for PV solar energy and (2) land for crops and for wind energy and summarized and mapped global results.

## Data availability
Input datasets and key spatial maps generated in this study have been deposited in Figshare at https://figshare.com/articles/software/BalancingLandUse_Files_zip/26133865.

## Code availability
The code used to run the prioritizations analyses is available on Figshare: https://figshare.com/articles/software/BalancingLandUse_Files_zip/26133865.

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

## Acknowledgements

We acknowledge funding from the UC Santa Barbara-Conservation International Climate Solutions Collaborative. A.E.F., B.J.E., and P.R.R.

acknowledge support from U.S. National Science Foundation Grant 2225076. C.M.K., J.R.O., and J.K. acknowledge support from The Nature Conservancy and One Earth. P.A.M. acknowledges support from Centro de Modelamiento Matemático (CMM), Grant FB210005, BASAL funds for Centers of excellence from ANID-Chile, the International Center for Theoretical Physics (ICTP) through their Associates Programme funded by the Simons Foundation through grant number 284558FY19, and the Earth Comission and Global Commos Alliance.

## Author contributions

L.H., P.R.R., A.E.L., and C.B. conceived the study. L.H., P.R.R., and A.E.L. acquired funding. The study concept and methodology were refined by T.B., B.J.E., A.E.F., J.A.J., C.M.K., A.E.L., R.L., P.A.M., R.A.N., A.R., R.S., D.R.W., G.C.W., and A.Z. C.B. led data collection and analysis with supervision by P.R.R. P.R.R., R.C.K., B.J.E., J.A.J., J.K., R.A.N., J.R.O., and L.H. provided data. C.B. drafted the initial text, and all authors reviewed and edited the manuscript.

## Competing interests

The authors declare no competing interests.

## Additional information

[1]Moore Center for Science and Solutions, Conservation International, Arlington, VA, USA. [2]Potsdam Institute for Climate Impact Research, Potsdam, Germany. [3]Global Science, WWF, San Francisco, CA, USA. [4]Department of Ecology and Evolutionary Biology, University of Arizona, Tucson, AZ, USA. [5]Department of Geography, University of California, Santa Barbara, CA, USA. [6]Department of Applied Economics, University of Minnesota, Saint Paul, MN, USA. [7]Global Science, The Nature Conservancy, Fort Collins, CO, USA. [8]Global Protect Oceans, Lands and Waters, The Nature Conservancy, Fort Collins, CO, USA. [9]Bren School of Environmental Science & Management, University of California, Santa Barbara, CA, USA. [10]Brazilian Foundation for Sustainable Development, Rio de Janeiro, RJ, Brazil. [11]Department of Ecology, Federal University of Goiás, Goiânia, GO, Brazil. [12]Facultad de Ciencias Biológicas, Pontificia Universidad Católica de Chile, Santiago, Chile. [13]The Santa Fe Institute, Santa Fe, NM, USA. [14]Centro de Cambio Global UC, Facultad de Ciencias Biológicas, Pontificia Universidad Católica de Chile, Santiago, Chile. [15]Instituto de Sistemas Complejos de Valparaíso (ISCV), Valparaíso, Chile. [16]Centro de Modelamiento Matemático (CMM), Universidad de Chile, International Research Laboratory 2807, CNRS, Santiago, Chile. [17]Wildlife Conservation Society, Center for Global Conservation, Bronx, NY, USA. [18]Department of Natural Resources and Environment, Cornell University, Ithaca, NY, USA. [19]Center for Natural Climate Solutions, Conservation International, Arlington, VA, USA. [20]Department of Biology, Carleton University, Ottawa, ON, Canada. [21]Nature Conservancy of Canada, Toronto, ON, Canada. [22]Sustainability Research Institute, School of Earth and Environment, University of Leeds, Leeds, UK. [23]Environmental Studies, University of California, Santa Barbara, CA, USA. ✉e-mail: cameryn.brock@gmail.com

