## [Transparent Peer Review File · Nature Communications]

Balancing land use for conservation, agriculture, and renewable energy

Corresponding Author: Ms Cameryn Brock

Version 0:

Reviewer comments:

Reviewer #1

(Remarks to the Author)

The authors make a case for multi-functional landscapes by demonstrating that such multi-sector planning can lead to beneficial reductions in number of threatened species and carbon loss, while without such multi-functional landscapes it is impossible to meet both conservation and development targets. I enjoyed reading the paper, and found the analysis to be novel and insightful. I recommend the paper be published with some modifications (however with the caveat that an unsuitable answer to comment 1 will lead to a change in my evaluation):

1. The authors mention `tradeoffs` multiple times in the paper, but do not show an actual tradeoff curve or pareto optimality frontier that can be traversed by a policy maker in order to identify the varied solution set and associated costs available to them. The cost in this case would be how many grid cells need to be transitioned from their current land use to a new one. Just because the integer-programming approach can find an optimal or near-optimal solution does not mean that the cost will not be so prohibitively large as to be impractical. If the authors analysis shows that the costs are impractical, then I will no longer be able to recommend publication.
2. I found figure 4 confusing, can you add what each color denotes specifically (the same way you did with black) in this figure?
3. In figure 2, most current cropland area has been transitioned to some other land-use without compensating in other regions. E.g. U.S Midwest seems to have no cropland; India has minimal cropland in a couple of states; No agriculture around the Nile etc. How is this realistic even in a future projection?
4. Going back to figure 4, what percentage of land under potential conflict is already belonging to the same land use class as either the Nature-first or the production first-scenario?
5. Does the `prioritizr` package allow clustering such that we do not get pixelated results i.e. one grid cell is conservation and the very next grid cell is development (e.g. a chess board pattern)? If so, how did the authors ensure contiguity of land use classes?

(Remarks on code availability)

The link did not work, i.e. I could not find any code

Reviewer #2

(Remarks to the Author)

This manuscript presents a framework for allocating land uses to mitigate threats to biodiversity, carbon, and ecosystem services while supporting development. The authors used a combination of linear programming, spatial prioritization, and scenarios to show how differing priorities affect optimal land allocation. The study highlights the need for collocation of

multiple land use sectors to meet conservation and development goals. The paper is novel, interesting, and ambitious, and should be of interest to a broad international community. I do have several questions that require better explanations and justifications, particularly on the methods, and on the spatial patterns in the maps (some of which were surprising, at least to me).

On the methods:

Please explain a little more the basis of mixed integer linear programming and how it works. Also, the citations in lines 90-92, used to justify the approach, are not in the references section.

Please explain the rationale for not allowing expansion in areas with human footprint value <0.2

Line 320 says "For some problems, it was not possible to meet development and/or conservation targets within the available land." Please explain more on how many countries were affected by this problem, if this issue may have resulted in inconsistencies in the results, and how these inconsistencies could have affected the comparisons and/or the overall findings.

Please provide more information on how species' exposure to development was measured, and on the magnitude of that exposure. My understanding is that if a species was projected to see development within its predicted habitat (i.e., habitat loss), then that species counts as impacted by development, correct? Then, does this mean that 1 pixel of habitat loss is enough to count the species as impacted by development? Is that ecologically meaningful? To get a better understanding on the impact of development on species, it would be useful to report not only how many species are impacted, but how large, on average, are those impacts (i.e. the proportion of the species' predicted habitat affected by development). That will give a better idea of the ecological meaning of the impact, as e.g. 5% habitat loss is not the same as e.g. 40% habitat loss.

Other comments:

Figure 4 identifies areas of potential conflict between conservation and development around the globe, representing a key result of this paper. Some of the spatial patterns, however, seem surprising, at least to me. For instance, the Chaco region, in Paraguay and northern Argentina, is known for being an important area for biodiversity and a global deforestation hotspot. Although the figure is a little blurry, I was surprised that this region appears with no conflicts between conservation and development. More surprising, agricultural expansion in Argentina is projected to happen far away in the drier, western region of Argentina, which is not the typical region for agriculture there. Thus, the question here is: how well is the model at allocating the land uses within each country? In other words, how do we know that land uses are projected in the right places? I understand that the aim of this paper is not to try to predict the future, but the model should be accurate at predicting whether/where two land uses compete for the same spot of land. Ultimately, the findings of this paper rely on how and where the model allocates land uses. More information would be helpful.

The text in pages 5 and 6 is not totally clear, with an excess of numbers reported, making it sometimes difficult to follow and retain the main messages.

Fig 2 shows land allocated for conservation and development in the multi-sector planning scenario. Please include the maps for the other scenarios as well, including under the two conservation options. These could be included in the appendix.

In the discussion, please explicitly answer the hypothesis stated in line 58.

Finally, this is an open question, but could the use of a different land use model approach provide a drastically different answer on what we need to meet development and conservation goals?

(Remarks on code availability)

Reviewer #3

(Remarks to the Author)

The abstract is concise and well-structured, but it should mention the method used. The reference to a framework is less sounding and generalist.

In my opinion, the manuscript's structure should be improved, placing the section on the methods after the main text. This led the reader to understand how scenarios framework has been developed, the tools used and the assumptions adopted.

Authors provide a framework based on an integrative spatial planning approach and ability to integrate multiple objectives and assess tradeoffs. The inclusion of multiple objectives in land-use planning is not new, but a certain noteworthy could be fine in how they combine this approach with country-level data, local and Indigenous perspectives (Heiner et al., 2019; Kennedy et al., 2023), and ongoing efforts to limit land conversion (such as through zero-deforestation commitments; Garrett et al., 2019).

In this sense, the study is relevant to the field, but its significance needs to be disclosed compared to established literature. Authors mention previous research shortly when should be more comprehensive. A more exhaustive description of the

literature is crucial to understand the novelty and justify some scientific assumptions as for example: why authors choose Nature's Contribution to People (NCP) and not Nature Futures Framework (NFF)? Authors adopts integer linear programming with the spatial conservation prioritization R package 'prioritizr. Many other tools could be used to develop conservation scenarios (i.e Zonation or MARXAN): why do R package 'prioritizr has been the choice? I would refer the same rationale of justify assumption to the choice of NCP. A consistent literature review exists on NCP but no citation about this topic. Showing that this study articulates NCP with prioritization R package 'prioritizr is relevant, even more when articulated to SSP framework.

Such reinforcement of the literature review and comparison with other study is strongly recommend also to improve the conclusions. Insights from this study are not only for land-use planning: they could be extremely relevant to orient Internacional and nacional policies. That aspect is missing in the texts. Indeed, conclusion are very poor and generalist and as such, need to be reformulated. In do that, I suggest authors to reflect on what stakeholders could be use the results from this study but also think about what the limitations of this approach (i.e resolution? Spatial scales? Others?). The calibration of a mathematical model and projections from a database could provide some results that are not real: for example, I am curious to ask authors what they think about the crops and food area in northern coastal Africa in Algeria and Morocco (fig.2) and its relationship with water scarcity, soil quality and population migrations. I suggest introducing some commentary about this point.

The following general typos have also to be reviewed:

The reference to Hansen et all, 2023 has to be review to Hanson et al, 2023.

Check all the references in the bibliography: many authors are on the list but are not referred to in the text.

I recommend authors to review the work to meet the expected standards of the journal and the field itself which is complex and relevant.

(Remarks on code availability)

This DOI cannot be found in the DOI System.

Version 1:

Reviewer comments:

Reviewer #2

(Remarks to the Author)

The authors did a great job at incorporating the comments, and I appreciate the time and effort at explaining an clarifying the information presented, as well as at expanding the caveats.

My only remaining (minor) comment, is that the results for the Chaco region are still surprising, as local studies projecting future land use changes for the Chaco region seem to show substantially higher rates of deforestation (and thus, potential conflicts with conservation) compared to this study (e.g. <https://doi.org/10.1007/s10113-022-01965-5>; <https://doi.org/10.1088/1748-9326/ad44b6>). At a discretion of the authors, I do wonder if it is worth adding in the limitations that locally calibrated land use models may give very different results, at least for some regions.

(Remarks on code availability)

Reviewer #4

(Remarks to the Author)

The approach of the study to compare global "production-first" and "nature-first" land use maps, and then try to find an "optimal" map combining both targets, is original and a meaningful contribution to literature. While I recognise the many assumption choices that need to be made in such an exercise, I would need to flag several issues I encounter in the methodology and the related conclusions. While there are several minor points, my main worry is that the different land uses in the study (productive and natural ones) are assumed to be entirely exclusive, while current reality already shows that is often not the case.

See below my chronological comments on the received manuscript.

Introduction:

- While the rest of the document nicely outlines specific impacts for different energy uses (wind, solar, water, energy crops), the second paragraph of the introduction piles up the literature on this topic in one sentence (lines 37-39 "Land use .. with agriculture"). While I understand that the format in Nature journals require only a short introduction, the land-use reality, and also existing literature, is quite varying for each variable. The same sentence could be more distinguished by renewable energy source, providing the appropriate literature for each.

Scenario framework:

- line 79, "within ... 2050": Why SSP1? SSP1 projections for food are significantly lower than those of other SSPs, as behavioural change is assumed to lower meat and therefore crop demand. Also, the sentence lets the reader think that SSP1 is also used for the energy demand projections, but the Methods indicate that Jacobson et al (2019) is used for this.
- line 84, "(for example, ... lands)": This example is precisely one of the combinations that is quite viable and already happening currently. Only the base of the tower, which is an insignificant amount of land, is exclusive of other uses.

Results:

- line 136-137, "76% ... turbines)": Again, this is a huge area in which the spacing areas can easily be used for other purposes, like agriculture and solar, and to a certain extent even natural purposes. In Figure 2, is this entire blue area in Northeastern China assumed to be for wind power only?
- line 171, "approximately ... achieved": Important doubt: since hydropower is assumed to follow its "current market share" in the calculated targets, according the Methods, fulfilling only half of this target could mean that current hydropower is being removed. This would most likely not make sense from an overall sustainability perspective, as hydro reservoirs can also be used for natural purposes, and the main ecological damage from creating a hydro reservoir is probably at the time of construction, hence reverting the land use may be hardly valuable (and even worse, if hydropower is allocated elsewhere as a result). This also leads to the overall question of how the model treats areas that already have renewables installed nowadays, are these current installations entirely subject to change in your methodology?
- Fig 3, panel B: Using stacked columns is not meaningful for efficiency values.
- Lines 221-222, "The ASIA ... by wind": This is a crucial part of the results as it strengthens my point above that wind power is quite compatible with other land uses, which apparently would quite strongly affect the overall results of the study and hence the conclusions.
- lines 249-250, "land allocated .. areas (35%)": Again an important question here is whether current hydropower is included in this figure (see also next comment). If so, that would confirm that conservation areas are compatible with hydro reservoirs in the real world.
- Fig 5: While the upper panel figure says to only refer to additional production, it is not clear whether current production adds to the overlap in the lower panel.

Discussion & conclusions:

- line 271, (Ortiz et al, 2022): There is likely much more evidence for this claim, certainly in the realm of bioenergy, even between those references you already cited in the introduction.
- lines 275-276, "While our .. Ven et al., 2021)": It would be good if this claim could be backed up by some data, or even a table making such comparisons with IAM outcomes.
- lines 287-289, "Multi-functional .. improve yields": I see this important critic on my end coming back as a policy recommendation here. While it indeed could be a meaningful recommendation for land-based solar power, which is hardly combined with other land uses nowadays, wind power is already consistently co-located with other land uses nowadays, so more than an outcome from the paper, I see it as a lack in the methodology.
- lines 342-343, "Our framework .. energy technologies": Honestly surprised by this statement. I would say that conventional energy technologies (conventional is commonly including fossil and nuclear power plants) are much more flexible in terms of siting, as they are not bound to meteorological circumstances.

Methods:

- lines 422-424, "This distinction .. reduction targets": While I understand the concept of robustness in this method choice, what if these impacts do not occur, as the renewable energy targets are attempting to accomplish? Wouldn't the allocation of crop production be significantly different? Probably worth doing such a sensitivity.
- line 426, "land for pasture is not considered": And what happens with this enormous share of land? Is it assumed to stay constant, or deemed available or other land uses? Currently unclear in manuscript.
- line 466, "future .. scenario": Also here, it would be useful to see if allocation of natural areas would drastically change when using a climate scenario more aligned with the renewable energy objectives followed.

(Remarks on code availability)

Reviewer #5

(Remarks to the Author)
Please see attached document.

(Remarks on code availability)

Version 3:

Reviewer comments:

Reviewer #4

(Remarks to the Author)

While the authors have adapted the manuscript in several places to address my comments, have have unfortunately chosen to not adapt their methodology and the analysis itself, despite the clear flaws layed out in the previous review round. The paper primarily focuses on land allocation between different future uses, though they still ignore the fact that certain land uses can be, and typically are, co-located in the same area. This flaw unfortunately has a significant impact on their final results and message, meaning that I can not support publication in a prestigious journal like Nature Communications in its

current form.

Specifically, the authors keep arguing against allowing coexistence of wind power and agricultural land (the large majority of today's wind power is installed in cropland or pasture; Maguire et al, 2024) in their methodology. Specifically, they seem to give a lot of weight Trainor et al (2016) in their argument, a relatively old paper in Plos One (IF~3) that introduces the concept of "land-scape level impact" as an alternative to the direct area footprint. While I do not argue in that wind power has wider landscape impacts, the focus of the paper is clearly on land allocation, and the wind that blows over that land is indeed important for wind power, but does not prevent it from being co-located with at least agricultural land, and also to a certain extent natural land types (while I understand the constraints, and the context-specific nature of the co-location with natural land). The authors also mention "limitations in available global data and the challenges of adequately capturing tradeoffs to biodiversity and ecosystem function in multi-use landscapes" in their argument, which is understandable, but since the gross of the argument focuses on the landscape impact of wind, it is doubtful if indeed nothing could have been done methodologically (at least allowing for the obvious potential coexistence between wind power and agriculture) or whether the authors preferred not to re-run the analysis. Additionally, methodological constraints are not necessarily a reason to accept flawed results.

On the impact of the abovementioned flaw on the consequential anomalies in the results (i.e. huge areas for "only wind" in China and the USA), the authors provide the answer: "Our global map is intended to highlight global patterns and areas of potential convergence in land-use demands; their purpose is not to prescribe precise local land-use decisions but to inform where more detailed, context-specific planning and coordination will be most critical. The wind production areas in northeastern China and the central United States are prime examples of where there is a need to integrate knowledge across scales". The issue is that in the context of integrating wind power with agriculture, this need does not really exist, as it is already common practice. While I understand it is not focused on prescribing precise local land use decisions, the authors do take some important conclusions from it, such as "Our results indicate that if the increase in development to meet future demand relies solely on additional land conversion, it is not possible to meet the country-level targets for conservation and development needs", a conclusion which may be affected for many countries when properly reflecting potentials for co-locating land uses. Also the results in Figure 5 are an important output of the paper, and strongly affected by the choice of not accounting for co-locating at least wind and agricultural land, which could free up significant land areas and reduce overlap.

Finally, while the authors seem well up-to-date on the literature on land conservation, the newly added sentences in the introduction alleviate they are less informed about the latest research on the energy-land nexus (only a few and very dated papers are cited), which is a critical topic in the context of their paper. For the purpose of the introduction itself, but also for avoiding obvious flaws in the results, I recommend the authors to dig a bit deeper into more recent papers about land use competition and co-existence of renewable energy with other land uses.

(Remarks on code availability)

Reviewer #5

(Remarks to the Author)

The authors have done a good job in addressing the issues raised. I do not wish to raise further comments at this stage.

(Remarks on code availability)

Version 4:

Reviewer comments:

Reviewer #4

(Remarks to the Author)

The authors have done a proper job in recognising co-location of future development, which is relatively smaller than expected from their initial results. With these changes, I would accept publication in its current form.

(Remarks on code availability)

Response to Reviewer Comments

We appreciate all the reviewers' valuable suggestions and comments on our manuscript, they have been instrumental in improving and revising our work and we are greatly appreciative. We have carefully reviewed all suggestions and made corrections accordingly with point-by-point responses.

We would like to highlight the following overarching updates:

- We have evaluated and discussed tradeoffs, including with Pareto optimality (Figs. S7-8) and further examination of land efficiency under the scenarios (Table S1), based on feedback from Reviewer 1.
- We have expanded upon key analytical decisions in the Methods, including regarding usage of a boundary penalty and the human modification index based on feedback from Reviewers 1 and 2.
- We have reorganized and streamlined our results section (pages 5 and 6, specifically) to be clearer to the reader, based on feedback from Reviewer 2.
- We have analyzed and reported species-level exposure to land allocated for development under each scenario (Fig. S8) based on feedback from Reviewer 2.
- We have elaborated on the need for careful consideration of the scale of implementation of this framework based on feedback from Reviewers 2 and 3.
- We have clarified and elaborated on the methodology in the Main Text to improve accessibility and to better define the parameters of the analysis based on feedback from Reviewer 3.

In the following pages, the comments from reviewers are in blue italics and our responses are in black. Our resubmission package includes Manuscript files with and without tracked changes; please note that line numbers are referring to the clean manuscript which has revisions hidden.

Thank you for your invaluable feedback.

Reviewer #1

The authors make a case for multi-functional landscapes by demonstrating that such multi-sector planning can lead to beneficial reductions in number of threatened species and carbon loss, while without such multi-functional landscapes it is impossible to meet both conservation and development targets. I enjoyed reading the paper, and found the analysis to be novel and insightful. I recommend the paper be published with some modifications (however with the caveat that an unsuitable answer to comment 1 will lead to a change in my evaluation):

1. The authors mention `tradeoffs` multiple times in the paper, but do not show an actual tradeoff curve or pareto optimality frontier that can be traversed by a policy maker in order to identify the varied solution set and associated costs available to them. The cost in this case would be how many grid cells need to be transitioned from their current land use to a new one. Just because the integer-programming approach can find an optimal or near-optimal solution does not mean that the cost will not be so prohibitively large as to be impractical. If the authors analysis shows that the costs are impractical, then I will no longer be able to recommend publication.

We appreciate this suggestion from the reviewer and have addressed the concept of trade-offs in several ways.

First, we have taken the reviewer's suggestion and created Pareto frontiers that illustrate a comparison between production output (gigawatts of renewable energy or tonnes dry matter from cropland) and conservation targets met per unit of cost (100km² of land), which we have described in the Main Text and added to the supplementary information. While the conditions in each country are different, the figures provide a visual comparison of the severity of tradeoffs across countries.

Below is Fig. S7, which reflects the Pareto frontier reflecting countries' individual optimizations for the countries with the highest area of overlapping priorities for conservation and production (i.e., countries identified in Fig. S10 right panel). Several of the countries that are highlighted in the frontier illustrate trade-offs between scenarios, even when all scenarios are highly effective. For example, in the right panel, all three scenarios for Indonesia are highly effective, with the Production-First planning scenario producing higher returns for cropland but lower for conservation targets, Nature-First enabling higher returns for conservation but lower for production, and Multi-Sector in between. We introduce these figures and have elaborated further on tradeoffs in multiple ways in lines 180-192.

Fig S7. Pareto optimality frontier comparing production output (renewable energy in left panel, cropland in right panel) and conservation targets met per 100km² for countries with the highest area of overlapping priorities for conservation and production (i.e., countries identified in Fig S10 right panel). Color depicts scenario; all scenarios reflect those run without a constraint on land allocated for conservation.

Second, we have reported the amount of land allocated per target met. The exact costs of allocation per unit area will vary, potentially by orders of magnitude, between and within countries depending on a multitude of variables outside our scope. Therefore, we believe that reporting the amount of land allocated provides the best summary metric for this kind of global analysis. These values are reported both in text (lines 187-188) and in Table S1. This is in addition to the sector-specific reporting of land used per unit of production in Figure 3B. We find that the total land allocated per sector varied across scenarios: mean land allocated ranged from 5,134-5,172 km² per target met for conservation and from 7,154-8,254 km² for production, with the Multi-Sector scenario falling in between the other scenarios for both. We believe this range demonstrates that in an average case, cost across scenarios should not be overly prohibitive. These results vary by country: the Supplementary Data details the area and percentage of land allocated per country per scenario.

Table S1. Additional land allocated for conservation and production per target met under each scenario. Information reflects scenarios run with a 30% constraint on land allocated for conservation.

Scenario	Mean Additional Land Allocated for Conservation per Conservation Target Met	Mean Additional Land Allocated for Production per Production Target Met
Production-First	5,171.60 km ²	7,154.78 km ²
Multi-Sector	5,170.49 km ²	8,005.35 km ²
Nature-First	5,134.41 km ²	8,254.85 km ²

2. I found figure 4 confusing, can you add what each color denotes specifically (the same way you did with black) in this figure?

We have added additional descriptors to reflect all the colors in Figure 4 to the figure caption. We have also added a small note with arrows to the figure itself to highlight which categories reflect areas in which conservation and development overlap.

3. In figure 2, most current cropland area has been transitioned to some other land-use without compensating in other regions. E.g. U.S Midwest seems to have no cropland; India has minimal cropland in a couple of states; No agriculture around the Nile etc. How is this realistic even in a future projection?

The results from the prioritizations are all additive, meaning they reflect the additional development beyond current land use needed to meet additional demand between our baseline and 2050. This means for the areas that the reviewer mentioned, Figure 2 shows what additional cropland is allocated beyond the current large cropland areas. Land transitions from other development are not permitted: we 'lock out', or omit, currently highly modified land from what land is available to the prioritization algorithm. This methodological decision is intended to highlight the land demands needed if future demand is solely met by conversion to production landscapes rather than with co-location of multiple sectors. We have clarified that we are referring to additional land allocated in figure captions for Figures 2, 3, 4, and 5.

4. Going back to figure 4, what percentage of land under potential conflict is already belonging to the same land use class as either the Nature-first or the production first-scenario?

We believe the reviewer is asking how much land under potential conflict has a current land use of currently developed (production) or protected (conservation) land. If that is correct, we would like to clarify that we have excluded areas of high human modification from what land is available to the prioritization, meaning that we do not allocate land, for conservation or production, to areas that are already highly modified. This is based on a human modification (HM) index value of 0.8 or higher (Methods follow Johnson et al., 2020). We acknowledge that this does not exclude all landscapes with some element of current production, particularly with consideration of aggregation to 5-km pixels, but we consider it the most defensible threshold for excluding highly modified landscapes in the literature. Regarding areas that are already protected for conservation, we do allow allocation of production on those landscapes, but with an additional cost (see Methods, lines 439-443), to signal potential risk to biodiversity and NCP in those areas. Fig S11 highlights areas where we see substantial overlap between allocation for land for development and current conservation areas, with notable areas including Mongolia (7,200 km²), China (5,150 km²), Japan (5,025 km²), Italy (4,975 km²), and Niger (4,150 km²) (as described in lines 242-43).

5. Does the `prioritizr` package allow clustering such that we do not get pixelated results i.e. one grid cell is conservation and the very next grid cell is development (e.g. a chess board pattern)? If so, how did the authors ensure contiguity of land use classes?

We appreciate the opportunity to expand on this analytical decision. Prioritizr does allow clustering. The primary method to do so across multiple sectors, and likely the most widely used method in general, is with a 'boundary penalty'. We considered whether to use this approach and opted not to for several reasons. First, the appropriate penalty value varies by prioritization. Since we ran prioritizations at the country level, we could not determine a consistent approach for determining a value across countries. In early runs of the prioritizations, we did test different boundary penalties, and found the results to be highly sensitive to the value selected and to be inconsistent in the level of clustering across countries. Once boundary penalties are introduced, the subjectivity of a solution increases substantially, potentially negating the gains one gets from using an optimization approach. Second, recent literature suggests that, for conservation, this approach can

lead to solutions that are highly dissimilar to others and may result in solutions that don't effectively provide functional connectivity (Hanson et al., 2022). Thus, we opted against introducing this potential source of bias in the results. Lastly, we are working with a pixel-based approach in which the entire 5km² is allocated at once, ensuring at least 5km² of contiguity with any land allocation. For context, over 80% of the terrestrial protected areas in the World Database on Protected Areas (WDPA) are smaller than 5km² (UNEP-WCMC & IUCN, 2021). Given the other caveats of using the boundary penalty approach, we felt this was sufficient. Despite the lack of explicit consideration of connectivity in our analysis, we did not find our results to be overly pixelated, likely due to the nature of the underlying data (e.g., agricultural suitability and development potential for renewables tend to be 'clustered' themselves).

We have added language to the Methods (lines 378-382) to reflect this decision.

Reviewer #1 (Remarks on code availability):

The link did not work, i.e. I could not find any code

We apologize that it appears the link to code was not working. We have fixed this issue. The code and data are available on figshare: https://figshare.com/articles/software/BalancingLandUse_Files_zip/26133865.

Reviewer #2

This manuscript presents a framework for allocating land uses to mitigate threats to biodiversity, carbon, and ecosystem services while supporting development. The authors used a combination of linear programming, spatial prioritization, and scenarios to show how differing priorities affect optimal land allocation. The study highlights the need for colocation of multiple land use sectors to meet conservation and development goals. The paper is novel, interesting, and ambitious, and should be of interest to a broad international community. I do have several questions that require better explanations and justifications, particularly on the methods, and on the spatial patterns in the maps (some of which were surprising, at least to me).

On the methods:

Please explain a little more the basis of mixed integer linear programming and how it works.

We have provided more detail on mixed integer linear programming to lines 100-105.

Also, the citations in lines 90-92, used to justify the approach, are not in the references section.

We thank the reviewer for catching our mistakes in the citations. We have added Schuster et al., 2020 to the references, and fixed a misspelling of Hanson (originally, 'Hansen'). We have double-checked all citations to ensure the reference list accurately reflects what is in the text.

Please explain the rationale for not allowing expansion in areas with human footprint value <0.2

Because we were evaluating the amount of land needed for development without co-location of multiple development sectors, we omitted land that currently has extensive human modification from the land that was available. We used the human modification index (HM; Kennedy et al., 2019) to determine HM. We excluded areas with a more than 80% degree of human modification following methods in Johnson et al., 2020. Previously, this was worded as 0.2 after subtracting the HM from 1: while technically true, we recognize this language was unnecessarily confusing and we have revised the text to be more clear. We have also added additional text for clarification in the methods (lines 367-372). We acknowledge that this does not exclude all landscapes with some element of current production, particularly with consideration of aggregation to 5 km² pixels, but we felt it was the most defensible threshold for creating a layer that accurately reflected available land in the literature.

Line 320 says "For some problems, it was not possible to meet development and/or conservation targets within the available land." Please explain more on how many countries were affected by this problem, if this issue may have resulted in inconsistencies in the results, and how these inconsistencies could have affected the comparisons and/or the overall findings.

We have added details on how many countries were affected in lines 373-374 (7 out of 160 countries). All results are evaluated against the original targets, so the results reflect the fact that not all countries can meet their targets. For example, in theory, if all countries were able to meet their targets within available land, Figure 3A would show 100% for percent of development targets met in the Production-First Planning Scenario and 100% for percent of conservation targets met in the Nature-First Planning Scenario (with no land constraint). Instead, we see 94.9% and 96.0-99.6%, respectively, due to some countries not being able to meet targets within available land. We clarify this in text in lines 376-378.

Please provide more information on how species' exposure to development was measured, and on the magnitude of that exposure. My understanding is that if a species was projected to see development within its predicted habitat (i.e., habitat loss), then that species counts as impacted by development, correct? Then, does this mean that 1 pixel of habitat loss is enough to count the species as impacted by development? Is that ecologically meaningful? To get a better understanding on the impact of development on species, it would be useful to report not only how many species are impacted, but how large, on average, are those impacts (i.e. the proportion of the species' predicted habitat affected by development). That will give a better idea of the ecological meaning of the impact, as e.g. 5% habitat loss is not the same as e.g. 40% habitat loss.

We agree with the reviewer that solely basing impact to a species from development by any overlap may not be meaningful. Instead, we defined species exposure based on the proportion of each species' range overlapping with the development: specifically, a species was impacted if at least 10% of its habitat intersected with the development footprint. We have added this clarification to lines 182-83, and this is restated in the caption of Figure 3 (line 210). We also accepted the reviewer's suggestion to report the proportion of each species range affected by development: we have added a supplementary Figure S8, below, that reflects this per scenario and per IUCN Red List category, and added a summary of these results to lines 195-197.

Fig S8. Percent of each species' habitat exposed to development across scenarios. In the box plots, the center lines denote median values, boxes extend from the 25th to the 75th percentile of each group's distribution, and the whiskers extend to a maximum of 1.5 times the interquartile range.

Other comments:

Figure 4 identifies areas of potential conflict between conservation and development around the globe, representing a key result of this paper. Some of the spatial patterns, however, seem surprising, at least to me.

For instance, the Chaco region, in Paraguay and northern Argentina, is known for being an important area for biodiversity and a global deforestation hotspot. Although the figure is a little blurry, I was surprised that this region appears with no conflicts between conservation and development. More surprising, agricultural expansion in Argentina is projected to happen far away in the drier, western region of Argentina, which is not the typical region for agriculture there. Thus, the question here is: how well is the model at allocating the land uses within each country? In other words, how do we know that land uses are projected in the right places? I understand that the aim of this paper is not to try to predict the future, but the model should be accurate at predicting whether/where two land uses compete for the same spot of land. Ultimately, the findings of this paper rely on how and where the model allocates land uses. More information would be helpful.

We recognize that the maps are difficult to examine closely in the manuscript and we have now provided spatial files for readers to review more closely with the publication. In the example of the Chaco region, based on boundaries from the WWF ecoregions dataset, there are nearly 5,000 km² of conflict area, visualized below. Further, there are overlaps of developmental priority areas with current conservation areas for Reserva Biologica Isla Yacreta in southern Paraguay and Humedales Chaco along northeastern Argentina. Because we designed current protected areas to have a higher cost in the prioritization than other areas, these overlaps indicate high development pressures. The spatial data are provided with the code so that readers can dig into areas of interest more effectively.

[Figure Redacted]

Map from Figure S11 (similar to Figure 4 with the addition of the red category which highlights overlap of additional land allocated for production with current protected areas), zoomed into the Chaco region.

Regarding western Argentina, we have reviewed our results compared to current land use and see that our land allocation for agriculture in the area all expands from areas of stable and expanding agriculture based on global data from Potapov et al., 2022. Our allocation of cropland includes consideration of how suitability will change under climate change, which may be what is driving a shift in agriculture from where it is typically found. Indeed, western Argentina is indicated as a climate-driven agricultural frontier, meaning an area that

becomes newly suitable for one or more crops based on climate change models (Hannah et al., 2020). Agricultural expansion in northwestern Argentina is also deemed as potential from an econometric perspective (Piquer-Rodríguez et al., 2018).

While we felt it was important to address the reviewer's comment regarding specific areas, we recognize the larger question of how well the model is at allocating land uses within each country. However, we are not aiming to identify exactly where conflicts will occur in the future; rather, our goal is to identify regions for which conflict may be high to prompt the need for country-specific assessment with more spatially accurate national and regional datasets. We also aim to highlight a larger issue: while the exact areas of conflict may not be those we identify, the fact that we have identified such large regions strongly implies that competition for land across conservation and development sectors will be a large problem in the future that we argue 1) should be focused on and 2) can be mitigated through strategic planning. What this paper allows is the initiation of a debate about these issues at a national or regional level, so we can integrate knowledge across scales (Johnson et al., 2023; Chaplin-Kramer et al., 2021).

The text in pages 5 and 6 is not totally clear, with an excess of numbers reported, making it sometimes difficult to follow and retain the main messages.

We have substantially reorganized and streamlined the text in pages 5 and 6, removing numbers where possible, with the intention of making the main messages clearer for the reader. We thank the reviewer for this suggestion as we feel it has greatly improved the flow and interpretability of our results.

Fig 2 shows land allocated for conservation and development in the multi-sector planning scenario. Please include the maps for the other scenarios as well, including under the two conservation options. These could be included in the appendix.

We have added additional figures to the Supplementary Information that reflect land allocated in the other scenarios (see Figs. S2-3). For the scenarios that do not have the 30% of land restraint applied to land allocation for conservation, Fig. S4 reflects the conservation solution for the Nature-First planning scenario. The development solution for the Production-First scenario will be nearly identical to that in Fig. S2 (methodologically they were assessed in the same way, the only small differences would be due to randomness in running the prioritization solver at different times): for this reason, we have opted not to include another map to avoid confusion. For assessing what conservation targets are technically possible to meet with remaining land in the Production-First scenario (and the reverse: production targets in the Nature-First scenario; see lines 409-413), all land is allocated, so visualizing those maps is not meaningful.

In the discussion, please explicitly answer the hypothesis stated in line 58.

We have added text to the discussion to answer our hypothesis, including explicitly doing so in lines 272-274.

Finally, this is an open question, but could the use of a different land use model approach provide a drastically different answer on what we need to meet development and conservation goals?

We appreciated the opportunity to reflect on this question. Ultimately, we believe that using different land use modeling approaches would likely provide different predictions in terms of the extent and spatial distribution of development, however we suspect that our high-level finding, that there will be conflicts between land use, that multi-sector planning will mitigate conflicts, and that there is insufficient remaining land to achieve conservation and development goals if we don't plan carefully, would all hold. This

optimization approach is based on a suite of assumptions and changing those assumptions or switching to another methodology (e.g., an empirical statistical model) would change the results.

We have higher confidence in the inputs we use for conservation and biophysical suitability, and believe optimal locations for these should remain relatively stable across data sources and models. Conversely, our model is also dependent on data for demand for production, which is highly subjective and defies long-term prediction. This means that the conflict areas themselves should not differ substantially between optimization-based land-use models, but the severity of conflict may vary more depending on demand.

A key variable that the model is sensitive to is that of the scale at which we allocate demand: doing so at the country-level differs greatly from that of the continent-level, and, in the other direction, at the sub-regional level. In the United States, for example, states with Democratic majorities tend to engage in more climate policy activity like incentivizing renewable energy development, whereas anti-climate policy activity such as subsidizing fossil fuel industries is more likely to occur in states with Republican leadership (Bromley-Trujillo & Holman, 2020). These state-level politics will drive the spatial patterns of future renewable energy development (and perhaps conservation) in a way that is not directly proportional to the resource potential, but this was not possible to consider in the model at our global- or country-level scale. It is possible that for countries like the United States for which regional governance varies substantially, our maps may be worse in ‘predicting’ development than in countries that have more homogenous governance. This is one of the reasons we emphasize that we do not recommend using our global maps for local planning, but instead encourage building local maps with our framework and integrating knowledge across scales. We have elaborated on this point further in the Discussion (lines 316-328).

There is a larger question here of the validity of a global model which is sensitive to the scale at which it is applied. This brings us back to our point in response to the focus on the Chaco region, above: The purpose of this paper is not to definitively state exactly where conflict will occur, but rather to highlight that conflicts are possible; to demonstrate that taking a multi-sectoral approach is likely to avoid these conflicts better than focusing on one sector alone; and to provide a solid base for more nuanced, smaller-scale discussions. We do not believe that these core conclusions will vary with a different modeling approach.

Reviewer #3

The abstract is concise and well-structured, but it should mention the method used. The reference to a framework is less sounding and generalist.

We have revised the abstract to include the method used.

In my opinion, the manuscript's structure should be improved, placing the section on the methods after the main text. This led the reader to understand how scenarios framework has been developed, the tools used and the assumptions adopted.

We followed the journal guidelines which (to our knowledge) require us to include the Methods at the end, and we appreciate that this format aims to be more inclusive to a general audience. We understand the reviewer's concern that more information is needed up front: to address this, we have provided a more detailed summary of the approach in the main text (lines 85-108), while still aiming to avoid redundancy where possible.

Authors provide a framework based on an integrative spatial planning approach and ability to integrate multiple objectives and assess tradeoffs. The inclusion of multiple objectives in land-use planning is not new, but a certain noteworthy could be fine in how they combine this approach with country-level data, local and Indigenous perspectives (Heiner et al., 2019; Kennedy et al., 2023), and ongoing efforts to limit land conversion (such as through zero-deforestation commitments; Garrett et al., 2019).

In this sense, the study is relevant to the field, but its significance needs to be disclosed compared to established literature. Authors mention previous research shortly when should be more comprehensive. A more exhaustive description of the literature is crucial to understand the novelty and justify some scientific assumptions as for example: why authors choose Nature's Contribution to People (NCP) and not Nature Futures Framework (NFF)? Authors adopts integer linear programming with the spatial conservation prioritization R package 'prioritizr. Many other tools could be used to develop conservation scenarios (i.e Zonation or MARXAN): why do R package 'prioritizr has been the choice?

I would refer the same rationale of justify assumption to the choice of NCP. A consistent literature review exists on NCP but no citation about this topic. Showing that this study articulates NCP with prioritization R package 'prioritizr is relevant, even more when articulated to SSP framework.

We thank the reviewer for these comments. We agree with the importance using this framework in combination with location-specific data and knowledge, in partnership with Indigenous peoples and local communities, to integrate knowledge across scales and to identify appropriate interventions that align with national and global goals. This point is stated in the text in lines 333-335.

Regarding the use of NCP: We have added context to the main text regarding our use of NCP, including a brief description: NCP builds on the ecosystem services concept to recognize the role of culture in the linkage between people and nature (Diaz et al., 2018). We have operationalized NCP in this study with the use of spatially explicit data that combine ecological supply of benefits with the populations they benefit (Chaplin-Kramer et al., 2022); to the best of our knowledge, spatially-explicit data are not available for NFF. The use of spatial data for NCP allows us to build on previous studies like Johnson et al., 2020 and Neugarten et al., 2024 to identify where conservation and development goals conflict. Further, NCP has special relevance as it has been explicitly stated in important international policy frameworks including the UN CBD GBF and IPBES. The added additional specification regarding NCP is in the main text in lines 94-97.

Regarding the use of prioritizr: Mixed integer linear programming solvers can outperform other approaches in reaching cost-effective solutions for conservation problems, both in terms of the optimality of solutions and in reducing computational power (Beyer et al., 2016; Schuster et al., 2020). Because our global-scale optimization included over 7,000 species alongside additional goals, prioritizr stood out as both computationally efficient and, when combined with Gurobi, sufficiently powerful to solve large optimization problems. We have provided more context regarding our decision to use prioritizr in lines 100-105.

Such reinforcement of the literature review and comparison with other study is strongly recommend also to improve the conclusions. Insights from this study are not only for land-use planning: they could be extremely relevant to orient Internacional and nacional policies. That aspect is missing in the texts. Indeed, conclusion are very poor and generalist and as such, need to be reformulated.

We agree with the reviewer and appreciate their perspective on the relevance of these results for international and national policies. For example, in the United States, these types of studies have been used to inform renewable energy transmission planning, data have been provided to project developers so that they may avoid high impact areas, government entities may pursue zoning programs to incentivize development in low-impact areas by preemptively planning transmission access to those areas or providing expedited and lower cost permitting. We have added more specificity of who can benefit from using this framework to the conclusion (lines 344-346) in place of general language we had previously. We have also aimed to add additional references to relevant literature throughout the Introduction.

In do that, I suggest authors to reflect on what stakeholders could be use the results from this study but also think about what the limitations of this approach (i.e resolution? Spatial scales? Others?).

We agree with the reviewer that there are limitations to this approach and we have addressed some of the larger limitations in the discussion (see lines 305-328, with lines 316-328 specific to scale and resolution). We have similarly addressed the topic of the scale of implementation in our response to Reviewer 2's last question, and we refer the reviewer to that response.

Regarding resolution: We agree that resolution is a model decision which will influence the results. For example, Neugarten et al., 2024 found that more land area was required to achieve NCP targets, consistent with previous studies (e.g., Arponen et al., 2012). In addition to adding discussion of this topic in the Discussion in lines 305-328 referenced in the above paragraph, this consideration is also addressed in the Methods (lines 361-363). Our framework is deliberately flexible to handle variable spatial scale and resolution depending on each planner's unique context.

Additional limitations are addressed in our Discussion, including the impact of shifting climate patterns on renewable energy capacity, uncertainty regarding future technology developments, and lack of consideration of additional sectors that may drive land expansion outside of renewable energy and agriculture. We have also added consideration of some of the conservation features we have left out, such as insects, plants, freshwater biodiversity, and the lack of representation of all available NCP. We believe that this means our conservation targets are likely an under-representation of what is truly needed to meet biodiversity and climate goals.

The calibration of a mathematical model and projections from a database could provide some results that are not real: for example, I am curious to ask authors what they think about the crops and food area in northern coastal Africa in Algeria and Morocco (fig.2) and its relationship with water scarcity, soil quality and population migrations. I suggest introducing some commentary about this point.

With regard to our results highlighting potential land allocation for cropland in northern coastal Africa: Water scarcity, soil quality, and population migrations are variables which drive the underlying data we used. The data for future cropland expansion include projected regional demand under the SSP framework, which is driven by future population and urbanization projections, among other variables (Riahi et al., 2017); future country-level market share based on spatially-explicit projections which consider precipitation, topography, and soil quality (Chen et al., 2022), and projected crop suitability under climate change which similarly considers soil data alongside multiple climatic variables (Fischer et al., 2021). Areas of northern Algeria and Morocco similarly light up as a climate-driven agricultural frontier, meaning an area that becomes newly suitable for one or more crops based on climate change models (Hannah et al., 2020).

While the reviewer refers to a specific region, we believe this question addresses a similar point as Reviewer 2 in questioning the validity of a global model when looking at local areas. We point the Reviewer to that response as well (see first response under 'Other comments'). In summary, we agree that global models are not suitable for local land use planning, and we would like to emphasize that the purpose of this paper is not to definitively state exactly where future land uses will definitively occur, but instead to provide a solid base for more nuanced, smaller-scale discussions. Our aim is to enable a discussion about these issues at a national or regional level, so we can integrate knowledge across local, regional, and global scales. We have emphasized this point in the Discussion in lines 329-335.

The following general typos have also to be reviewed:

The reference to Hansen et all, 2023 has to be review to Hanson et all, 2023.

Check all the references in the bibliography: many authors are on the list but are not referred to in the text.

I recommend authors to review the work to meet the expected standards of the journal and the field itself which is complex and relevant.

We thank the reviewer for catching the referenced errors. All have been corrected.

Reviewer #3 (Remarks on code availability):

This DOI cannot be found in the DOI System.

We apologize that it appears the link to code was not working. We have fixed this issue. The code and data are available on figshare: https://figshare.com/articles/software/BalancingLandUse_Files_zip/26133865.

References

- Arponen, A., Lehtomäki, J., Leppänen, J., Tomppo, E., & Moilanen, A. (2012). Effects of Connectivity and Spatial Resolution of Analyses on Conservation Prioritization across Large Extents. *Conservation Biology*, 26(2), 294–304. <https://doi.org/10.1111/j.1523-1739.2011.01814.x>
- Bromley-Trujillo, R., & Holman, M. R. (2020). Climate Change Policymaking in the States: A View at 2020. *Publius: The Journal of Federalism*, 50(3), 446–472. <https://doi.org/10.1093/publius/pjaa008>
- Chaplin-Kramer, R., Brauman, K. A., Cavender-Bares, J., Díaz, S., Duarte, G. T., Enquist, B. J., Garibaldi, L. A., Geldmann, J., Halpern, B. S., Hertel, T. W., Khoury, C. K., Krieger, J. M., Lavorel, S., Mueller, T., Neugarten, R. A., Pinto-Ledezma, J., Polasky, S., Purvis, A., Reyes-García, V., ... Zafra-Calvo, N. (2021). Conservation needs to integrate knowledge across scales. *Nature Ecology & Evolution*. <https://doi.org/10.1038/s41559-021-01605-x>
- Chaplin-Kramer, R., Neugarten, R. A., Sharp, R. P., Collins, P. M., Polasky, S., Hole, D., Schuster, R., Strimas-Mackey, M., Mulligan, M., Brandon, C., Diaz, S., Fluet-Chouinard, E., Gorenflo, L. J., Johnson, J. A., Kennedy, C. M., Keys, P. W., Longley-Wood, K., McIntyre, P. B., Noon, M., ... Watson, R. A. (2022). Mapping the planet's critical natural assets. *Nature Ecology & Evolution*. <https://doi.org/10.1038/s41559-022-01934-5>
- Díaz, S., Pascual, U., Stenseke, M., Martín-López, B., Watson, R. T., Molnár, Z., Hill, R., Chan, K. M. A., Baste, I. A., Brauman, K. A., Polasky, S., Church, A., Lonsdale, M., Larigauderie, A., Leadley, P. W., Van Oudenhoven, A. P. E., Van Der Plaats, F., Schröter, M., Lavorel, S., ... Shirayama, Y. (2018). Assessing nature's contributions to people. *Science*, 359(6373), 270–272. <https://doi.org/10.1126/science.aap8826>
- Fischer, G., Nachtergaele, F.O., van Velthuizen, H.T., Chiozza, F., Franceschini, G., Henry, M., Muchoney, D. and Tram-berend, S. (2021). Global Agro-Ecological Zones v4 – Model documentation. Rome, FAO. <https://doi.org/10.4060/cb4744e>
- Hannah, L., Roehrdanz, P. R., K. C., K. B., Fraser, E. D. G., Donatti, C. I., Saenz, L., Wright, T. M., Hijmans, R. J., Mulligan, M., Berg, A., & van Soesbergen, A. (2020). The environmental consequences of climate-driven agricultural frontiers. *PLOS ONE*, 15(2), e0228305. <https://doi.org/10.1371/journal.pone.0228305>
- Hanson, J. O., Vincent, J., Schuster, R., Fahrig, L., Brennan, A., Martin, A. E., Hughes, J. S., Pither, R., & Bennett, J. R. (2022). A comparison of approaches for including connectivity in systematic conservation planning. *Journal of Applied Ecology*, 1365-2664.14251. <https://doi.org/10.1111/1365-2664.14251>
- Johnson, J. A., Kennedy, C. M., Oakleaf, J. R., Baruch-Mordo, S., Polasky, S., & Kiesecker, J. (2021). Energy matters: Mitigating the impacts of future land expansion will require managing energy and extractive footprints. *Ecological Economics*, 187, 107106. <https://doi.org/10.1016/j.ecolecon.2021.107106>
- Johnson, J. A., Brown, M. E., Corong, E., Dietrich, J. P., Henry, R. C., Jeetze, P. J. von, Leclère, D., Popp, A., Thakrar, S. K., & Williams, D. R. (2023). The meso scale as a frontier in interdisciplinary modeling of sustainability from local to global scales. *Environmental Research Letters*, 18(2), 025007. <https://doi.org/10.1088/1748-9326/acb503>
- Molotoks, A., Stehfest, E., Doelman, J., Albanito, F., Fitton, N., Dawson, T. P., & Smith, P. (2018). Global projections of future cropland expansion to 2050 and direct impacts on biodiversity and carbon storage. *Global Change Biology*, 24(12), 5895–5908. <https://doi.org/10.1111/gcb.14459>
- Neugarten, R. A., Chaplin-Kramer, R., Sharp, R. P., Schuster, R., Strimas-Mackey, M., Roehrdanz, P. R., Mulligan, M., Van Soesbergen, A., Hole, D., Kennedy, C. M., Oakleaf, J. R., Johnson, J. A., Kiesecker, J., Polasky, S., Hanson, J. O., & Rodewald, A. D. (2024). Mapping the planet's critical areas for biodiversity and nature's contributions to people. *Nature Communications*, 15(1), 261. <https://doi.org/10.1038/s41467-023-43832-9>

Piquer-Rodríguez, M., Baumann, M., Butsic, V., Gasparri, H. I., Gavier-Pizarro, G., Volante, J. N., Müller, D., & Kuemmerle, T. (2018). The potential impact of economic policies on future land-use conversions in Argentina. *Land Use Policy*, 79, 57–67. <https://doi.org/10.1016/j.landusepol.2018.07.039>

Potapov, P., Turubanova, S., Hansen, M. C., Tyukavina, A., Zalles, V., Khan, A., Song, X.-P., Pickens, A., Shen, Q., & Cortez, J. (2022). Global maps of cropland extent and change show accelerated cropland expansion in the twenty-first century. *Nature Food*, 3(1), 19–28. <https://doi.org/10.1038/s43016-021-00429-z>

Riahi, K., van Vuuren, D. P., Kriegler, E., Edmonds, J., O'Neill, B. C., Fujimori, S., Bauer, N., Calvin, K., Dellink, R., Fricko, O., Lutz, W., Popp, A., Cuaresma, J. C., Kc, S., Leimbach, M., Jiang, L., Kram, T., Rao, S., Emmerling, J., ... Tavoni, M. (2017). The Shared Socioeconomic Pathways and their energy, land use, and greenhouse gas emissions implications: An overview. *Global Environmental Change*, 42, 153–168. <https://doi.org/10.1016/j.gloenvcha.2016.05.009>

UNEP-WCMC and IUCN. (2021). Protected Planet: The World Database on Protected Areas (WDPA). <https://www.protectedplanet.net>

Response to Reviewers

Reviewer #2 (Remarks to the Author):

The authors did a great job at incorporating the comments, and I appreciate the time and effort at explaining and clarifying the information presented, as well as at expanding the caveats.

Response: We thank the reviewer for their constructive feedback, and we are pleased to hear we have addressed the queries raised to their satisfaction.

My only remaining (minor) comment, is that the results for the Chaco region are still surprising, as local studies projecting future land use changes for the Chaco region seem to show substantially higher rates of deforestation (and thus, potential conflicts with conservation) compared to this study (e.g. <https://doi.org/10.1007/s10113-022-01965-5>; <https://doi.org/10.1088/1748-9326/ad44b6>). At a discretion of the authors, I do wonder if it is worth adding in the limitations that locally calibrated land use models may give very different results, at least for some regions.

Response: We agree with the reviewer that localized land-use models, such as those applied to the Chaco region, can provide important insights that differ from global projections, particularly due to their ability to incorporate region-specific drivers, land-use dynamics, and policy contexts.

Our analysis is based on global-scale development potential indices and demand projections, which provide a consistent basis for cross-sector and cross-region comparisons. However, we recognize that this approach may underrepresent dynamics in regions like the Chaco where local pressures on land conversion are more acute than what global datasets capture.

In response to the reviewer's suggestion, we have added elaboration on this point to the discussion (lines 348-352 in clean MS) acknowledging that locally calibrated models may yield different results and can offer more accurate projections in certain regions, including specific reference to the Chaco region. We also note that integrating local modeling efforts with global assessments represents an important direction for future work and could improve the accuracy and applicability of spatial planning tools such as ours.

Reviewer #4 (Remarks to the Author):

The approach of the study to compare global "production-first" and "nature-first" land use maps, and then try to find an "optimal" map combining both targets, is original and a meaningful contribution to literature. While I recognise the many assumption choices that need to be made in such an exercise, I would need to flag several issues I encounter in the methodology and the related conclusions. While there are several minor points, my main worry is that the different land uses in the study (productive and natural ones) are assumed to be entirely exclusive, while current reality already shows that is often not the case.

Response: We appreciate the reviewer's assessment of the study as an original and meaningful contribution to the literature. We thank them for raising the important concern in our decision to treat land uses as mutually exclusive within a given planning unit. We agree that co-location of some sectors (e.g., in some contexts, co-locating wind or solar energy with agriculture) represents an important opportunity to reduce land demand and mitigate land-use conflicts, and we believe that integrating co-location more explicitly represents a valuable extension for future research. To address this point (and several similar points from the reviewer below), we have added elaboration to acknowledge this more clearly in the discussion (lines 304-314; all line numbers reference clean MS) and highlighted this in the introduction (lines 90-93). Further, we have added emphasis that the "conflict areas" identified by our study are not inherently incompatible for multiple uses but represent planning units where sectoral demands converge and warrant more detailed, landscape-level assessment (lines 234-237). We provide additional clarification below.

The choice to treat land uses as exclusive in our modeling framework was driven by several factors. First, although some co-location scenarios show promise, such as solar and grazing lands (Sharpe et al., 2021) and wind and agriculture (Maguire et al., 2024), we were unable to include these in our modeling framework given a lack of sufficient global data available regarding land suitability for mixed use landscapes specifically. For example, the Oakleaf et al. development potential models used in this study are designed to reflect industrial-scale land uses rather than diversified or subsistence-level systems. Even when land uses are theoretically compatible, many practical, context-specific considerations make successful co-locations complex and uncertain. These factors, along with local ecological sensitivities and management practices, are difficult to generalize or incorporate at a global scale, and the data and modeling tools currently available reflect a more conventional form of land development. While we recognize and specifically address in the manuscript the promise of co-location for some sectors (e.g., lines 299-304), we felt these issues presented a practical constraint to embedding assumptions about multi-use compatibility into our model.

Second, while there is technical evidence that energy infrastructure can be compatible with agricultural uses (Maguire et al., 2024; Ravi et al., 2016; Miskin et al., 2019), there is less research available addressing whether co-located land uses between development and natural lands support biodiversity and NCP. The limited literature that is available on biodiversity outcomes in shared-use landscapes suggests that biodiversity can fare poorly (Rehling et al., 2023; Phalan et al., 2011); this also aligns with theoretical work arguing that win-win outcomes

from land sharing are rare and that trade-offs are common (Hegwood et al., 2022). For example, while development for wind energy may avoid total conversion of land, this same pattern can drive habitat fragmentation and associated species impacts when sited in natural areas (Rehling et al., 2023; Kiesecker, 2024). Further, in addition to their contributions to habitat loss, wind and solar energy can cause direct mortality to some species, such as through collisions with wind turbines or solar panels or, in the case of concentrated solar energy, through burning (Kumara et al., 2022; Conkling et al., 2022). Because of these uncertainties and trade-offs, we believe it would be premature and potentially risky to assume co-location as a strategy counting toward conservation goals within a global spatial optimization framework.

See below my chronological comments on the received manuscript.

Introduction:

- While the rest of the document nicely outlines specific impacts for different energy uses (wind, solar, water, energy crops), the second paragraph of the introduction piles up the literature on this topic in one sentence (lines 37-39 "Land use .. with agriculture"). While I understand that the format in Nature journals require only a short introduction, the land-use reality, and also existing literature, is quite varying for each variable. The same sentence could be more distinguished by renewable energy source, providing the appropriate literature for each.

Response: We thank the reviewer for this feedback. We have further expounded in the second paragraph (specifically, lines 39-46), specifying the additional land needs of some renewable energy sources over others; for example, specifying the large amount of land needed in total and per unit of power of bioenergy when compared to other sources. As the projected large energy footprint is constituted of renewable and extractive energy sources, we have also added additional context comparing the future land use needs of each.

Scenario framework:

- line 79, "within ... 2050": Why SSP1? SSP1 projections for food are significantly lower than those of other SSPs, as behavioural change is assumed to lower meat and therefore crop demand.

Response: We appreciate the reviewer's question regarding our selection of the SSP 1 pathway. Our modeling approach integrates outputs from multiple datasets aligned with the SSP framework in a way that seeks to balance ambition and caution. Specifically, we based our land demand projections for renewable energy and food production on SSP 1, while modeling climate-driven shifts in species and crop distributions under a moderate climate change scenario. This distinction is intentional; it allows us to assess what would be required to meet sustainability-aligned development targets while also accounting for the potential ecological consequences of less optimistic climate outcomes.

Our choice of SSP 1 for land demand aligned with our objective to explore multi-sector land allocation under a scenario that aligns with the most ambitious global sustainability goals – those that seek to simultaneously mitigate climate change, reduce biodiversity loss, and promote responsible consumption. SSP 1 is consistent with a future in which strong land-use governance, rapid decarbonization, and widespread environmental awareness are realized, and where global development pathways are intentionally designed to reduce pressures on natural systems.

We acknowledge that SSP 1 food demand projections are lower than those in other SSPs due to several factors, including the assumed shifts in dietary patterns that the reviewer mentions. This aspect of SSP 1 reflects exactly the kind of behavioral and policy change that is increasingly promoted in global sustainability discussions (e.g., IPCC, CBD, and FAO pathways). Our aim is not to predict the most likely future, but rather to evaluate spatial risks and tradeoffs under a future that is aligned with international climate and biodiversity commitments. This decision is highlighted in the introduction in lines 133-136, but recognizing that clarification may be helpful when we first introduce the decision to use SSP 1, we have also added more detail in the sentence the reviewer highlighted (lines 85-87). Lastly, we have highlighted this decision and potential consequences for the amount of land allocated to development in the discussion (lines 339-341).

Also, the sentence lets the reader think that SSP1 is also used for the energy demand projections, but the Methods indicate that Jacobson et al (2019) is used for this.

Response: The introduction correctly states that SSP 1 is used for energy demand projections. The reviewer is correct that Jacobson et al., 2019 is involved as well: the distinction is that while SSP demand is reported at the regional level, this analysis allocates land at the country level. Thus, we needed projected market share at the country-level, for which we used Jacobson et al (2019). This is described in lines 456-460 and more briefly in the introduction in lines 87-88.

- line 84, "(for example, ... lands)": This example is precisely one of the combinations that is quite viable and already happening currently. Only the base of the tower, which is an insignificant amount of land, is exclusive of other uses.

Response: We thank the reviewer for highlighting this example. While we acknowledge that in some cases wind turbines can be physically co-located with other land uses such as agriculture, our modeling approach treats land uses as mutually exclusive due to a broader set of considerations. This is not intended to imply that technical co-location is impossible, but rather reflects limitations in available global data and the challenges of adequately capturing trade-offs to biodiversity and ecosystem function in multi-use landscapes. These limitations are discussed in the response to the initial reviewer comment above. We have added an additional sentence after the one the reviewer highlighted to clarify our decision (lines 92-93).

To address wind power specifically, even when the direct land take of wind energy infrastructure is minimal, the broader landscape impacts can be substantial. The installation

and ongoing maintenance of wind farms require access roads, expended infrastructure, and operational zones that alter habitat quality and increase fragmentation. Wind turbines also pose unique risks to biodiversity through direct mortality to birds and bats, which can be ecologically significant. Accordingly, we consider the entire planning unit where wind energy is allocated as affected by development. This framing is supported by recent work that shows that while the direct footprint of wind energy infrastructure is small, its landscape-level impact is approximately 100 times larger (Trainor et al., 2016). Their analysis found that wind has one of lowest land-use efficiencies when broader ecological and spatial impacts are considered, second only to bioenergy.

Results:

- line 136-137, "76% ... turbines)": Again, this is a huge area in which the spacing areas can easily be used for other purposes, like agriculture and solar, and to a certain extent even natural purposes. In Figure 2, is this entire blue area in Northeastern China assumed to be for wind power only?

Response: Please see the above responses related to our decision to treat land uses as exclusive and to consider spacing in between turbines as development area. The reviewer is correct that our model allocates the space in northeastern China for wind energy. Our global map is intended to highlight global patterns and areas of potential convergence in land-use demands; their purpose is not to prescribe precise local land-use decisions but to inform where more detailed, context-specific planning and coordination will be most critical. The wind production areas in northeastern China and the central United States are prime examples of where there is a need to integrate knowledge across scales (Chaplin-Kramer et al., 2021). Global maps such as these identify priority areas, trends, and connections while coordinated local actions can advance sustainable landscape planning (Frazier, 2024).

- line 171, "approximately ... achieved": Important doubt: since hydropower is assumed to follow its "current market share" in the calculated targets, according the Methods, fulfilling only half of this target could mean that current hydropower is being removed. This would most likely not make sense from an overall sustainability perspective, as hydro reservoirs can also be used for natural purposes, and the main ecological damage from creating a hydro reservoir is probably at the time of construction, hence reverting the land use may be hardly valuable (and even worse, if hydropower is allocated elsewhere as a result). This also leads to the overall question of how the model treats areas that already have renewables installed nowadays, are these current installations entirely subject to change in your methodology?

Response: All of the land use allocations that result from the optimization methods and form the basis for the results are *additional* installations of, for instance, wind energy, solar energy, cropland, etc. Any current land uses that are dedicated to those purposes and reflect high human modification have been locked out of the model and are not available for additional allocation of productive land uses and/or natural lands. These existing land uses contribute food, energy, etc. to meeting future projected needs, but all additional need beyond what can

be produced on the existing land must be met with new land. This means in our methodological approach it is not possible to have less land allocated for any of the sectors than what is currently allocated. We have aimed to make this clear to the reader by specifying “additional land allocated” throughout.

- Fig 3, panel B: Using stacked columns is not meaningful for efficiency values.

Response: We thank the reviewer for pointing this out. We recognize and agree that stacked columns are not the optimal way of presenting this information. As such we have modified Figure 3 to the below:

- Lines 221-222, "The ASIA ... by wind": This is a crucial part of the results as it strengthens my point above that wind power is quite compatible with other land uses, which apparently would quite strongly affect the overall results of the study and hence the conclusions.

Response: Please see the above responses related to our decision to treat land uses as exclusive and to consider spacing in between turbines as development area. The line the reviewer is referencing states there is substantial overlap between conservation and development lands, which is a key finding in this paper. We realize maybe this sentence could have been mistakenly interpreted to mean that wind and food crops were co-located (rather than discussing both their overlap with conservation lands), and we have clarified the sentence (lines 228-231). We have also added additional justification to the end of the paragraph (lines 234-237).

- lines 249-250, "land allocated .. areas (35%)": Again an important question here is whether current hydropower is included in this figure (see also next comment). If so, that would confirm that conservation areas are compatible with hydro reservoirs in the real world.

Response: See response to the following comment.

- Fig 5: While the upper panel figure says to only refer to additional production, it is not clear whether current production adds to the overlap in the lower panel.

Response: See response above clarifying that all land allocated is additional land, this relates to all figures and results reported. The numbers in Figure 5 are only for additional land allocation needed to meet future targets for each land use. Existing modified lands are "locked out" of land available for allocation in the model and are not represented here. We have added language in the figure caption to emphasize this point (line 269).

Discussion & conclusions:

- line 271, (Ortiz et al, 2022): There is likely much more evidence for this claim, certainly in the realm of bioenergy, even between those references you already cited in the introduction.

Response: We agree with the reviewer and thank them for this comment, specifically in the suggestion to highlight bioenergy here. We have added four additional citations (line 282-283) to provide support for this claim:

Fargione, J., Hill, J., Tilman, D., Polasky, S., & Hawthorne, P. (2008). Land Clearing and the Biofuel Carbon Debt. *Science*, 319(5867), 1235–1238. <https://doi.org/10.1126/science.1152747>

Keles, D., Choumert-Nkolo, J., Combes Motel, P., & Nazindigouba Kéré, E. (2018). Does the expansion of biofuels encroach on the forest? *Journal of Forest Economics*, 33, 75–82. <https://doi.org/10.1016/j.jfe.2018.11.001>

Rehling, F., Delius, A., Ellerbrok, J., Farwig, N., & Peter, F. (2023). Wind turbines in managed forests partially displace common birds. *Journal of Environmental Management*, 328, 116968. <https://doi.org/10.1016/j.jenvman.2022.116968>

Zhang, P., Yue, C., Li, Y., Tang, X., Liu, B., Xu, M., Wang, M., & Wang, L. (2024). Revisiting the land use conflicts between forests and solar farms through energy efficiency. *Journal of Cleaner Production*, 434, 139958. <https://doi.org/10.1016/j.jclepro.2023.139958>

- lines 275-276, "While our .. Ven et al., 2021)": It would be good if this claim could be backed up by some data, or even a table making such comparisons with IAM outcomes.

Response: We appreciate the reviewer's suggestion to clarify the relationship between our results and IAM outcomes. While our aim is not to predict precise land allocations, our methodological approach aligns with previous research that uses the SSP database to allocate land for future energy and food production (e.g., Popp et al., 2017; Johnson et al., 2021). We have updated the manuscript to clarify this (lines 287-288).

- lines 287-289, "Multi-functional .. improve yields": I see this important critic on my end coming back as a policy recommendation here. While it indeed could be a meaningful recommendation for land-based solar power, which is hardly combined with other land uses nowadays, wind power is already consistently co-located with other land uses nowadays, so more than an outcome from the paper, I see it as a lack in the methodology.

Response: We appreciate this perspective from the reviewer and have further emphasized the limitations of this methodological approach, as discussed in multiple of the responses above.

- lines 342-343, "Our framework .. energy technologies": Honestly surprised by this statement. I would say that conventional energy technologies (conventional is commonly including fossil and nuclear power plants) are much more flexible in terms of siting, as they are not bound to meteorological circumstances.

Response: We appreciate the reviewer's insight and can see that, in some respects, fossil and nuclear power plants can be more flexible in siting due to their lower dependence on local meteorological conditions. Our statement was intended to refer specifically to the broader land-use implications of sourcing energy, including the upstream infrastructure required to support sustained energy production. While the siting of a conventional power plant may be flexible, fossil fuel systems are inherently tied to the geographic location of extractive resources (e.g., coal seams, oil fields) and require continued land disturbance as new sources are developed over time (e.g., see Trainor et al., 2016). In contrast, renewable energy technologies – particularly wind and solar – can, in principle, be sited on already converted or disturbed lands and sustained over the long term without the need to continually open new areas. We have clarified this point in the revised manuscript to better reflect this nuance (line 373).

Methods:

- lines 422-424, "This distinction .. reduction targets": While I understand the concept of robustness in this method choice, what if these impacts do not occur, as the renewable energy targets are attempting to accomplish? Wouldn't the allocation of crop production be significantly different? Probably worth doing such a sensitivity.

Response: Our intention in making this distinction was to emphasize a robust planning approach that anticipates potential constraints on land productivity due to climate impacts, while also aiming for mitigation consistent with sustainability goals. We agree that if climate impacts are less severe than the scenario suggests, patterns of agricultural suitability could shift, potentially altering crop allocation outcomes. Exploring the sensitivity of results to alternative climate scenarios would be a valuable extension of this work in future research. We have added an additional sentence to highlight that the extent of future climate change will alter patterns of agricultural suitability (line 328).

- line 426, "land for pasture is not considered": And what happens with this enormous share of land? Is it assumed to stay constant, or deemed available for other land uses? Currently unclear in manuscript.

Response: To constrain what land was available for development, we used the human modification index (HMI; Kennedy et al., 2019), excluding all areas with more than 80% degree of human modification following Johnson et al., 2021. This means that what would happen with pasture depends on whether that area is deemed to have high human modification or not. If it is captured as >80% in the HMI layer, then that area is excluded from potential allocation for any category in the model. If it is not within that threshold, then that land is deemed as available for allocation for conservation or development in the model. We clarified the sentence the reviewer highlighted to make it clear that we meant that land for pasture is not considered as a development category (lines 455-456).

- line 466, "future .. scenario": Also here, it would be useful to see if allocation of natural areas would drastically change when using a climate scenario more aligned with the renewable energy objectives followed.

Response: We appreciate the reviewer's suggestion and agree that future work could explore how biodiversity priority areas may shift under different climate scenarios. In this study, we used an RCP 7.0 scenario to conservatively identify areas of stable climatic suitability for species (i.e., potential climate refugia). While the extent of suitable habitat might expand or contract with alternative scenarios, the geographic distribution of climate refugia tends to be relatively robust, the core areas of natural value emphasized in our results are unlikely to change drastically in location and may be best revealed by more extreme scenarios. We have added to the following sentence to clarify this rationale in the manuscript (lines 497-499).

Reviewer #5 (Copied from attachment):

Review of “Balancing land use for conservation, agriculture, and renewable energy”

A very nicely presented paper on an important topic. Clearly some considerable effort has also gone into responding to the reviewers comments. However, I did have some concerns.

I was surprised by how similar the maps of the 3 scenarios Production-First (Fig s2) Multi-Sector, and (Fig 2) Nature-First (fig S3) looked. Given the differences in the objectives in the optimisation I would have expected them to have looked much more different. I therefore looked at Figure 4 carefully as this seemed to show more conflict (i.e. variation).

For example, looking at one of the largest and most extreme conflicts (shown as black in Fig 4) of in the north of Libya. While in production-first there seems to be less conservation in that area than nature-first, it does not appear to have substantial development. Although the poor resolution of the maps in the SI does not help the comparison. Below are some extracts that show what I mean.

Extract from Fig 4: Conflict between production-first and nature-first

Extract from Fig S2. Production-first

Extract from Fig S3. Nature-first

Another example is north Algeria, where there are large orange areas (for food crop expansion) in both Figs S2 & 3, but is suggested to be a conflict in Fig 4.

The patterns of land expansion for cropland for food are not what are typically in most land use projections. These tend to use greater expansion tropical areas than shown here. To take the example of Algeria again. The results here have one of the largest cropland expansions there. However, this area already contains a lot of cropland, some land use projections suggest is one

of the areas most damaged by climate change and is abandoned from production in some higher warming RCPs. A dramatic expansion in that area is surprising for both those reasons.

Given the apparent inconsistency in results presented, (or perhaps my misunderstanding of the results). I therefore find it hard to have confidence in the analysis that is based on them.

Key questions are therefore:

1. Why are the patterns that arise from the different approaches so similar (very unexpectedly so to me).

Response: We thank the reviewer for their detailed feedback and careful assessment of the maps. Although the scenarios differ in whether they prioritize production or nature first, several factors lead to overlapping patterns. First, all scenarios draw from the same underlying data for development potential, which influence where agriculture and renewable energy are most viable. These layers that reflect biophysical and economic suitability constrain the optimization and tend to drive land allocation toward the same core areas of high potential, even when the order of objectives shift. Similarly for conservation outcomes, the underlying data driving the areas needed for biodiversity are the same across scenarios. Further, all scenarios maintain the same baseline constraints (e.g., excluding highly modified land, excluding or penalizing current conservation areas), and the same development targets for food and energy. While for different scenarios the objective changes which areas are conserved/developed first, the core development patterns trend similarly due to these shared constraints. We may expect the most pronounced differences to emerge in areas of marginal suitability or where conservation priorities overlap heavily with development potential, which are highlighted in the map of potential conflict areas (Figure 4).

2. Why are the 'conflicts' figure 4 not visible by a comparison of figures S2 and S3.

Response: We appreciate that the reviewer pointed out that it isn't clear what's all driving the Figure 4 map. The conflict map is derived from four scenario outputs, not just the two the reviewer referenced:

- 1) Production-first (30% land conservation target)
- 2) Production-first (no conservation constraint)
- 3) Nature-first (30% land conservation target)
- 4) Nature-first meet targets (no conservation constraint)

Three of these were previously included in the SI, but the Production-first (no conservation constraint) was not. We have now added this missing map to the SI to allow for full visual comparison (Fig. S4). Comparison of all four maps helps clarify why areas such as northern Libya and northern Algeria appear as zones of conflict in Figure 4: these regions are prioritized differently across these four scenarios. See these areas highlighted across all the maps in a visual comparison on the following page. We have also added new language to the Methods clarifying this process (lines 518-523 in clean MS).

REVIEWER COMMENTS

Reviewer #4 (Remarks to the Author):

While the authors have adapted the manuscript in several places to address my comments, have have unfortunately chosen to not adapt their methodology and the analysis itself, despite the clear flaws layed out in the previous review round. The paper primarily focuses on land allocation between different future uses, though they still ignore the fact that certain land uses can be, and typically are, co-located in the same area. This flaw unfortunately has a significant impact on their final results and message, meaning that I can not support publication in a prestigious journal like Nature Communications in its current form.

Response

We agree that co-location, especially between wind energy and cropland, is an important pathway to reduce land use. Our study's optimization intentionally used a single-use, pixel-based allocation because globally consistent data on the ecological and technical conditions needed to support sustainable co-location are not yet available (see lines 316-324; all line numbers reference clean manuscript).

However, to more directly address the reviewer's concern with regard to colocation of future cropland and future renewable energy production, we have added an analysis which compares independent allocations for select development sectors using our existing data and framework. We ran independent, single-use prioritizations for wind, photovoltaic (PV) solar, and cropland (includes both food and energy crops) and then analyzed the overlap among the land allocated for renewable energy and cropland. See the new Methods section, *Comparison of independent allocations for select development* on lines 534-539.

The additional analysis indicates that globally, 2.7% of land allocated for PV solar overlapped with land allocated for crops (or 1.1% of crop expansion overlapped with that for PV), primarily in Europe and India (new Fig. S10). 2.1% of land allocated for wind energy overlapped with that for crops (or 3.5% of crop expansion overlapped with that for wind), primarily in Europe and China, followed by eastern Brazil and Tanzania (new Fig. S11). These results highlight opportunities where co-location could reduce additional land conversion. We discuss these results as indicative of where co-location could reduce future land demand while noting that biodiversity outcomes and technical feasibility may vary. We summarize these results in the Results (lines 228-233) and now state upfront that our primary single-use allocation should be interpreted as an upper bound on land demands when co-location is not explicitly modeled for our climate and socio-economic scenario (lines 96-98).

Specifically, the authors keep arguing against allowing coexistence of wind power and agricultural land (the large majority of today's wind power is installed in cropland or pasture; Maguire et al, 2024) in their methodology. Specifically, they seem to give a lot of weight Trainor et al (2016) in their argument, a relatively old paper in Plos One (IF~3) that introduce the concept of "land-scape level impact" as an alternative to the direct area footprint. While I do not argue in that wind power has wider landscape impacts, the focus of the paper is clearly on land allocation, and the wind that blows over that land is indeed important for wind power, but does not prevent it from being co-located with at least agricultural land, and also to a certain extent natural land types (while I understand the constraints, and the context-specific nature of the co-location with natural land). The authors also mention "limitations in available global data and the challenges of adequately capturing tradeoffs to biodiversity and ecosystem function in multi-use landscapes" in their argument, which is understandable, but since the gross of the argument

focuses on the landscape impact of wind, it is doubtful if indeed nothing could have been done methodologically (at least allowing for the obvious potential coexistence between wind power and agriculture) or whether the authors preferred not to re-run the analysis. Additionally, methodological constraints are not necessarily a reason to accept flawed results.

Response

As introduced in our response to the previous comment, the new comparison of independent allocations analyses between wind and cropland and PV solar and cropland show that there are areas that may be well suited for both renewable energy and cropland, illustrating that if it were technically feasible for co-location to occur there, additional conversion could be reduced (see 228-233 and Figs. S10 and S11, described in the previous response). However, the potential is a small percentage of the total allocated land for wind power (2.1%), suggesting that endogenously including co-location within the model would only result in minor changes to the overall results. Additionally, see lines 307-313 in the Discussion in which we acknowledge co-location and cite work on wind-agriculture and solar-agriculture coexistence (e.g., Maguire et al., 2024; Ravi et al., 2016; Miskin et al., 2019; Deshmukh et al., 2019; Hernandez et al., 2019). The new analysis strengthens this point by identifying where multi-functional landscapes may be productive while emphasizing that technical feasibility of co-location and outcomes for biodiversity and NCP are not assessed.

On the impact of the abovementioned flaw on the consequential anomalies in the results (i.e. huge areas for "only wind" in China and the USA), the authors provide the answer: "Our global map is intended to highlight global patterns and areas of potential convergence in land-use demands; their purpose is not to prescribe precise local land-use decisions but to inform where more detailed, context-specific planning and coordination will be most critical. The wind production areas in northeastern China and the central United States are prime examples of where there is a need to integrate knowledge across scales". The issue is that in the context of integrating wind power with agriculture, this need does not really exist, as it is already common practice. While I understand it is not focused on prescribing precise local land use decisions, the authors do take some important conclusions from it, such as "Our results indicate that if the increase in development to meet future demand relies solely on additional land conversion, it is not possible to meet the country-level targets for conservation and development needs", a conclusion which may be affected for many countries when properly reflecting potentials for co-locating land uses. Also the results in Figure 5 are an important output of the paper, and strongly affected by the choice of not accounting for co-locating at least wind and agricultural land, which could free up significant land areas and reduce overlap.

Response

We thank the reviewer for their feedback. We would like to provide two clarifications. First, we do not necessarily limit development to only natural areas; rather, we only exclude highly modified areas with a human modification index of above 0.8 (see Methods lines 408-410). When allocating land, this methodology does allow for allocation on some agricultural land (e.g., wind in the U.S. Midwest) and thus assumes a degree of colocation of existing cropland with other sectors. Second, our results portray additional potential future allocation for each sector. This means the areas indicated as wind in the results reflect places selected to meet future demand under a single-use assumption; it not a claim that wind cannot co-locate with cropland or other sectors in those areas. While we acknowledge that wind and cropland can be (and often are) co-located, the expansion of each does not necessarily occur simultaneously.

We have added clarification to the way we discuss our results in specifying that we are assuming co-location does not occur in future expansion (lines 165, 307). We have adjusted the line the reviewer specifically mentions to “Our results indicate that if the increase in development to meet future demand relies solely on additional land conversion without co-locating sectors (e.g., wind energy on agricultural land), it is not possible to meet the country-level targets for conservation and development needs” (lines 164-166). We also added the following specification to the Fig. 1 caption: “Expansion of multiple sectors in one pixel was not considered” (line 127).

We agree that the magnitude of additional land implied by single-use allocation would decline where co-location is feasible, and this is highlighted in lines 17, 227-228, and 307-309. We have added specification that our single-use allocations represent an upper bound on land needs (lines 96-98) and the new comparison of independent allocations analyses results provide quantitative context.

Finally, while the authors seem well up-to-date on the literature on land conservation, the newly added sentences in the introduction alleviate they are less informed about the latest research on the energy-land nexus (only a few and very dated papers are cited), which is a critical topic in the context of their paper. For the purpose of the introduction itself, but also for avoiding obvious flaws in the results, I recommend the authors to dig a bit deeper into more recent papers about land use competition and co-existence of renewable energy with other land uses.

Response

We appreciate the suggestion to strengthen our coverage of recent energy-land nexus research and we agree that including more up-to-date references improves the manuscript. We have now expanded the Introduction to summarize current evidence on relative land use intensity of different energy technologies and the extent of land use conflicts with solar and wind energy, using global analysis where available (though, we acknowledge that most social science research is from the U.S.). We also reference studies that provide solutions to these conflicts through avoidance or dual use. These additions update our framing while remaining consistent with our global, single-use optimization scope.

Reviewer #5 (Remarks to the Author):

The authors have done a good job in addressing the issues raised. I do not wish to raise further comments at this stage.

Response

We thank the reviewer and greatly appreciate their feedback.

REVIEWER COMMENTS

Reviewer #4 (Remarks to the Author):

The authors have done a proper job in recognising co-location of future development, which is relatively smaller than expected from their initial results. With these changes, I would accept publication in its current form.

Response

We thank the reviewer for their helpful feedback.

Review of “Balancing land use for conservation, agriculture, and renewable energy”

A very nicely presented paper on an important topic. Clearly some considerable effort has also gone into responding to the reviewers comments. However, I did have some concerns.

I was surprised by how similar the maps of the 3 scenarios Production-First (Fig s2) Multi-Sector, and (Fig 2) Nature-First (fig S3) looked. Given the differences in the objectives in the optimisation I would have expected them to have looked much more different. I therefore looked at Figure 4 carefully as this seemed to show more conflict (i.e. variation).

For example, looking at one of the largest and most extreme conflicts (shown as black in Fig 4) of in the north of Libya. While in production-first there seems to be less conservation in that area than nature-first, it does not appear to have substantial development. Although the poor resolution of the maps in the SI does not help the comparison. Below are some extracts that show what I mean.

Extract from Fig 4: Conflict between production-first and nature-first

Extract from Fig S2. Production-first

Extract from Fig S3. Nature-first

Another example is north Algeria, where there are large orange areas (for food crop expansion) in both Figs S2 & 3, but is suggested to be a conflict in Fig 4.

The patterns of land expansion for cropland for food are not what are typically in most land use projections. These tend to use greater expansion tropical areas than shown here. To take the example of Algeria again. The results here have one of the largest cropland expansions there. However, this area already contains a lot of cropland, some land use projections suggest is one of the areas most damaged by climate change and

is abandoned from production in some higher warming RCPs. A dramatic expansion in that area is surprising for both those reasons.

Key questions are therefore:

1. Why are the patterns that arise from the different approaches so similar (very unexpectedly so to me).
2. Why are the 'conflicts' figure 4 not visible by a comparison of figures S2 and S3.

Given the apparent inconsistency in results presented, (or perhaps my misunderstanding of the results). I therefore find it hard to have confidence in the analysis that is based on them.